# SOXF factors regulate murine satellite cell self-renewal and function through inhibition of β-catenin activity

**Sonia Alonso-Martin[1,2,3‡]\*, Frédéric Auradé[4†], Despoina Mademtzoglou[1,2,3†], Anne Rochat[4], Peter S Zammit[5], Frédéric Relaix[1,2,3,6,7]\***

[1]Institut Mondor de Recherche Biomédicale, INSERM U955-E10, Créteil, France; [2]Université Paris Est, Faculté de Medecine, Créteil, France; [3]Ecole Nationale Veterinaire d'Alfort, Maison Alfort, France; [4]Sorbonne Université, INSERM U974, Center for Research in Myology, Paris, France; [5]Randall Centre for Cell and Molecular Biophysics, King's College London, London, United Kingdom; [6]Etablissement Français du Sang, Creteil, France; [7]APHP, Hopitaux UniversitairesHenri Mondor, Centre de Référence des Maladies Neuromusculaires GNMH, Créteil, France

**Abstract** Muscle satellite cells are the primary source of stem cells for postnatal skeletal muscle growth and regeneration. Understanding genetic control of satellite cell formation, maintenance, and acquisition of their stem cell properties is on-going, and we have identified SOXF (SOX7, SOX17, SOX18) transcriptional factors as being induced during satellite cell specification. We demonstrate that SOXF factors regulate satellite cell quiescence, self-renewal and differentiation. Moreover, ablation of *Sox17* in the muscle lineage impairs postnatal muscle growth and regeneration. We further determine that activities of SOX7, SOX17 and SOX18 overlap during muscle regeneration, with SOXF transcriptional activity requisite. Finally, we show that SOXF factors also control satellite cell expansion and renewal by directly inhibiting the output of β-catenin activity, including inhibition of *Ccnd1* and *Axin2*. Together, our findings identify a key regulatory function of SoxF genes in muscle stem cells via direct transcriptional control and interaction with canonical Wnt/β-catenin signaling.
DOI: https://doi.org/10.7554/eLife.26039.001

**\*For correspondence:**
alonsomartin.s@gmail.com (SA-M);
frelaix@gmail.com (FR)

[†]These authors contributed equally to this work

**Present address:** [‡]Tissue Regeneration Laboratory, Centro Nacional de Investigaciones Cardiovasculares, Madrid, Spain

**Competing interests:** The authors declare that no competing interests exist.

## Introduction

Maintenance, repair, and regeneration of adult tissues rely on a small population of stem cells, which are maintained by self-renewal and generate tissue-specific differentiated cell types (*Weissman, 2000*). Most adult stem cells are quiescent within their niche, dividing infrequently to generate both a copy of the stem cell and a rapidly cycling cell (*Barker et al., 2010*). These features make adult stem cells essential for either normal tissue homeostasis or repair/regeneration following damage (*Slack, 2000*). Hence, identification and manipulation of stem cells, including understanding mechanisms of cell fate decision and self-renewal, are essential to develop stem cell-based therapeutic strategies (*Relaix, 2006*).

Skeletal muscle contains a population of resident stem cells - termed satellite cells (*Katz, 1961*; *Mauro, 1961*). Around birth, fetal muscle progenitor cells adopt a satellite cell position, becoming embedded within the basal lamina in close contact to the muscle fibers (*Ontell and Kozeka, 1984*; *Relaix et al., 2005*). Importantly, during postnatal growth, the emerging satellite cells progressively enter quiescence, a molecular state poorly characterized in vivo. However, in response to injury or

disruption of the basal lamina, satellite cells are activated and proliferate to form myoblasts that either fuse to existing myofibers to repair, or fuse together to form multinucleated *de novo* myotubes for regeneration. Alternatively, a subset of satellite cells self-renews to maintain a residual pool of quiescent stem cells that has the capability of supporting additional rounds of growth and regeneration (*Zammit et al., 2006*). Satellite cells are indispensable for muscle recovery after injury, confirming their pivotal and non-redundant role as skeletal muscle stem cells (reviewed in *Relaix and Zammit, 2012*).

Many studies have demonstrated a balance between extrinsic cues and intracellular signaling pathways to preserve stem cell function, with Notch and Wnt signaling being of particular importance (*Brack and Rando, 2012*; *Dumont et al., 2015*). Wnt signaling has been extensively studied in satellite cells (*Brack et al., 2008*; *Kuang et al., 2008*). Whereas canonical Wnt signaling, implying β-catenin/TCF activation, is upregulated upon muscle regeneration and regulates satellite cell differentiation (*Otto et al., 2008*; *von Maltzahn et al., 2012*), non-canonical Wnt signaling (independent of β-catenin), mediates satellite cell self-renewal and muscle fiber growth (*Le Grand et al., 2009*; *von Maltzahn et al., 2012*). However, how Wnt signaling pathways interact with intrinsic transcriptional regulators remains unclear. Therefore, identifying the transcriptomic changes in muscle progenitors and satellite cells through development, growth and maturity is fundamental in order to build a comprehensive model of satellite cell formation and function (*Alonso-Martin et al., 2016*). Focusing on the important transition from developmental to postnatal myogenesis, we identified the SOXF family (SOX7, SOX17, SOX18) as potentially having a pivotal role in muscle stem cell function (*Alonso-Martin et al., 2016*).

SOX factors belong to the high mobility group (HMG) superfamily of transcription factors (*Bernard and Harley, 2010*), and act in the specification of stem cells in a number of tissues during development (*Irie et al., 2015*; *Lizama et al., 2015*). SOX17 plays important roles in development, particularly in embryonic stem cells (*Sarkar and Hochedlinger, 2013*; *Séguin et al., 2008*) and endoderm formation (*Hudson et al., 1997*; *Kanai et al., 1996*), and is critical for spermatogenesis (*Kanai et al., 1996*) and specification of human primordial germ cell fate (*Irie et al., 2015*). SOX17 is also implicated in stem cell homeostasis in adult hematopoietic tissues and in cancer (*Corada et al., 2013*; *He et al., 2011*; *Lange et al., 2009*; *Ye et al., 2011*). SOX7 shares a role in endoderm formation with SOX17, and interestingly, genetic interaction of *Sox7* with *Sox17* has been recently reported in developmental angiogenesis (*Kim et al., 2016*; *Shiozawa et al., 1996*; *Takash et al., 2001*). Finally, loss of SOX18 leads to cardiovascular and hair follicle defects (*Pennisi et al., 2000*). Moreover, SOX18 together with SOX7 and SOX17 regulates vascular development in the mouse retina (*Zhou et al., 2015*).

While SoxF genes play key functions in different stem cell systems, little is known of their role in myogenesis. Here, using a set of ex vivo and in vivo experiments including genetic ablation and regeneration studies, we demonstrate that these factors regulate skeletal muscle stem cell self-renewal as well as satellite cell-driven postnatal growth and muscle regeneration. Moreover, we show that SOXF factors operate via interaction with β-catenin in myogenic cells to modulate the output of Wnt canonical signaling during postnatal myogenesis.

## Results

### SoxF gene expression parallels satellite cell emergence and promotes satellite cell self-renewal

To characterize the formation, establishment and maintenance of satellite cells, we performed a chronological global transcriptomic profiling in embryonic, fetal, and postnatal muscle progenitors and satellite cells (*Alonso-Martin et al., 2016*). These cells were prospectively isolated from a $Pax3^{GFP/+}$ population, with minimal contamination of endothelial cells, as previously reported (*Alonso-Martin et al., 2016*) (*Figure 1—figure supplement 1*). Focusing on establishment of satellite cells, we identified the SOXF family (SOX7, SOX17, SOX18) of transcriptional regulators as likely key regulators of satellite cell function.

Strikingly, SoxF genes are barely detectable during embryonic and fetal stages (*Figure 1A–B*) but are induced at onset of the emergence of satellite cells and robustly expressed in postnatal satellite cells at the transcript and protein level (*Figure 1A–C*).

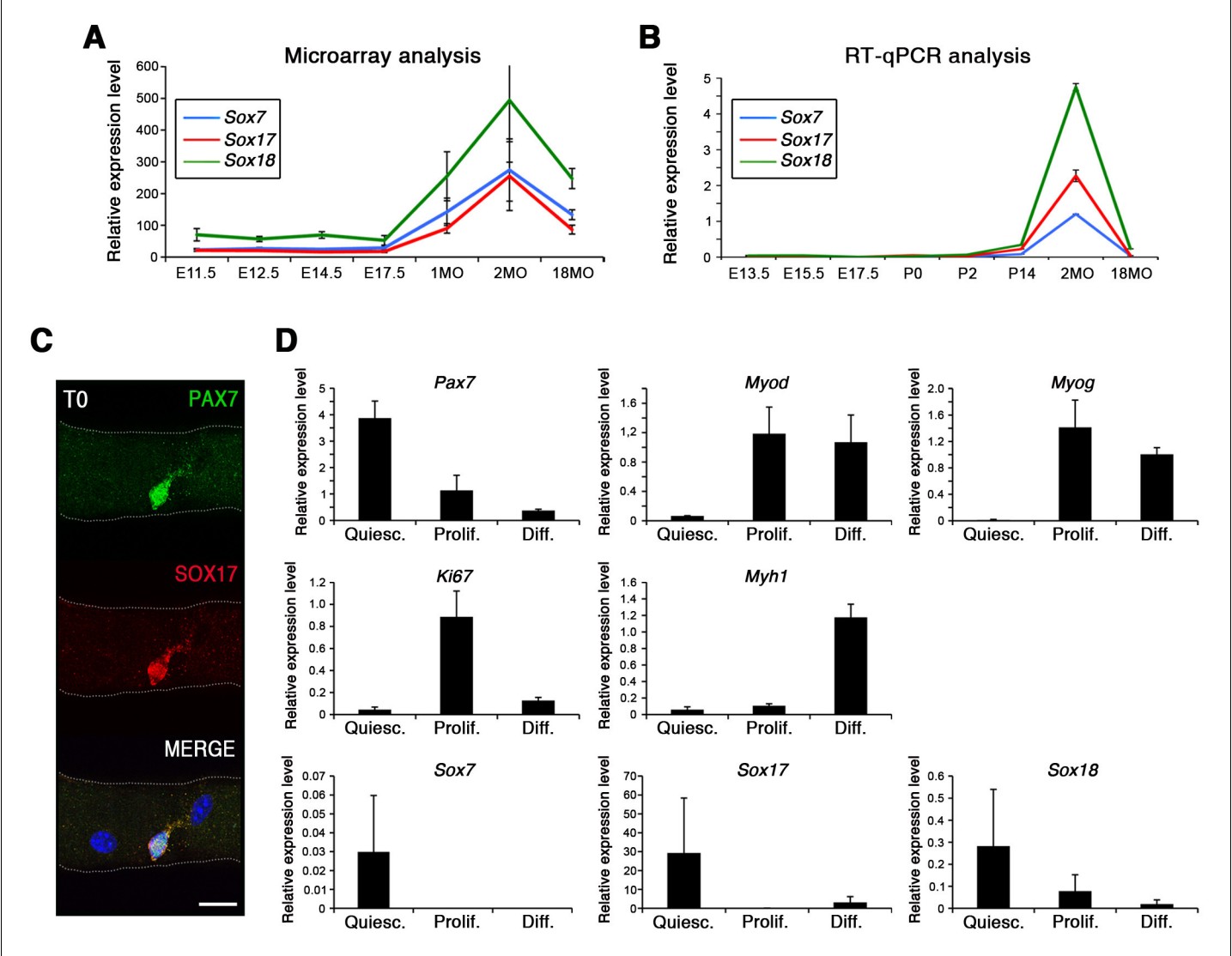

**Figure 1.** SoxF genes are induced at onset of satellite cell emergence and regulate adult myogenesis. (A,B) Expression levels of SoxF genes (*Sox7*, *Sox17*, *Sox18*) in FACS-isolated *Pax3^{GFP/+}* cells from Affymetrix expression analysis (A) and RT-qPCR (B). E, Embryonic day; P, Postnatal day; MO, age in months. (C) Representative immunolabeling of a satellite cell (PAX7+) co-expressing SOX17 on a freshly isolated adult myofiber (T0). Scale bar, 10 μm. Nuclei are counterstained with DAPI. (D) Expression profile of fresh FACS-sorted and cultured satellite cells for quiescence (*Pax7*), activation/commitment (*Myod*, *Myog*), proliferation (*Ki67*), terminal differentiation (*Myh1*), and for SoxF (*Sox7*, *Sox17*, *Sox18*) transcripts. Quiesc., quiescence; Prolif., proliferation; Diff., differentiation conditions. n = 3 mice (each quantified in triplicate) for all experiments. Data expressed as mean ± s.e.m.

DOI: https://doi.org/10.7554/eLife.26039.002

The following figure supplement is available for figure 1:

**Figure supplement 1.** Minimal CD31+ cell contamination in FACS-isolated skeletal muscle stem cells.

DOI: https://doi.org/10.7554/eLife.26039.003

To examine whether SOXF factors were present specifically in quiescent satellite cells, we performed primary culture experiments in proliferation and differentiation conditions. We isolated freshly FACS-sorted quiescent satellite cells and compared their expression profile to those undergoing culture (*Figure 1D*). Whereas activation (*Myod*), proliferation (*Ki67*), and differentiation (*Myog*, *Myh1*) transcripts were all induced in culture conditions, SoxF were predominately detectable in quiescent (*Pax7*) satellite cells (*Figure 1D*).

To characterize the role of SOXF factors in satellite cell function, we used the myofiber culture model, which maintains a functional niche for skeletal muscle stem cells while allowing their

observation (*Zammit et al., 2004*). We generated retroviruses encoding a bi-cistronic expression for full-length SOX7FL, SOX17FL or SOX18FL, or transactivation defective SOX7ΔCt, SOX17ΔCt or SOX18ΔCt proteins (*Figure 2—figure supplement 1A*), together with GFP to identify transduced cells. As SOXF proteins share the same consensus DNA binding sequence, any SOXFΔCt is expected to behave as a dominant negative for all three transcription factors (*Hou et al., 2017*). Retrovirus encoding IRES-GFP only was used as a control (CTRL). Overexpression of any of the SoxF genes (SOXF-FL) induced a similar phenotype in satellite cells, increasing the pool of self-renewing satellite cells (PAX7+/GFP+) (*Figure 2A–D*), concomitant with less activation (MYOD+/GFP+) (*Figure 2E–H*), proliferation (KI67+/GFP+) (*Figure 2I–L*), and differentiation (MYOG+/GFP+) (*Figure 2—figure supplement 1C–F*). All PAX7+/GFP+ cells underwent at least one division after exiting their quiescent state, as shown by EdU incorporation in transduced GFP+ cells (*Figure 2—figure supplement 1B*). This SOXF overexpression in satellite cells parallels the effects observed in other stem cell types, such as adult hematopoietic progenitors (*He et al., 2011*). Conversely, expression of transactivation defective SOXFΔCt caused a decrease in self-renewal (PAX7+/GFP+) (*Figure 2A–D*) and promoted proliferation (MYOD+/GFP+, KI67+/GFP+) of satellite cells (*Figure 2E–H and I–L*), but had no measurable effect on differentiation (MYOG+/GFP+) (*Figure 2—figure supplement 1C–F*). Taken together, these results show that SoxF genes promote self-renewal of adult muscle stem cells and their return to a mitotically quiescent state.

## SOX17 is required for satellite cell quiescence and myofiber maturation

Considering the important role of SOX17 in cell stemness and cell fate decisions (*Chhabra and Mikkola, 2011*; *Irie et al., 2015*; *McDonald et al., 2014*), we chose to investigate its function in postnatal skeletal muscle satellite cells in vivo. Since *Sox17* mutant mice die during development (*Kim et al., 2007*), we combined a null *Sox17* reporter allele (*Sox17^{GFP}*) with a conditional *Sox17^{fl}* allele to perform tissue-specific genetic ablation of *Sox17*: intercrossing with *Pax3^{Cre/+}* mice to achieve lineage-specific *Sox17* deletion during development and consequently postnatally, or *Pax7^{CreERT2/+}* mice for an inducible adult satellite-cell-specific deletion. *Pax3^{Cre/+};Sox17^{GFP/fl}* mutant mice had no obvious differences in body or muscle weight during postnatal growth or in adulthood (*Figure 3—figure supplement 1A–C*). Yet, *Sox17*-knockout *Soleus* muscle in adult *Pax3^{Cre/+};Sox17^{GFP/fl}* mice contained more myofibers, but with reduced cross-sectional area (*Figure 3A–D*). Myofibers from *Pax3^{Cre/+};Sox17^{GFP/fl}* *Soleus* also had a lower myonuclei density (*Figure 3E*), suggesting that *Sox17*-deficient muscles have less satellite cells contributing to postnatal muscle growth (*White et al., 2010*; *Yin et al., 2013*; *Zammit, 2008*). Indeed, direct quantification using PAX7 or MCAD immunolabeling, including reduction of *Pax7* transcripts, revealed that there were fewer satellite cells in *Pax3^{Cre/+};Sox17^{GFP/fl}* muscles (*Figure 4A,B,D* and *Figure 4—figure supplement 1*). Interestingly, this reduction was already evident by two weeks of postnatal growth (*Figure 4B*), a time when a significant proportion of satellite cells are becoming quiescent, forming the pool of adult muscle stem cells. Finally, consistent with our myofiber culture experiments (*Figure 2*), we found that the decrease in muscle stem cells in *Sox17*-knockout mice was associated with a striking decrease of quiescent cells (*Figure 4C*). Instead, an increased proportion of satellite cells expressed PAX7 and MYOD (18.3% vs. 3.4% in controls) in *Sox17*-knockout mutants, and thus were activated, and 16.8% even expressed just MYOD (compared to 2.4% in controls), indicating that they were potentially entering the differentiation program (*Figure 4C*).

Conditional knockout of *Sox17* specifically in adult satellite cells caused a similar loss of satellite cells as soon as three weeks after tamoxifen injection in *Pax7^{CreERT2/+};Sox17^{fl/fl}* mutant mice (*Figure 4E–G*). Myofiber content and morphology was not affected in satellite-cell-specific *Sox17*-conditional knockout (*Pax7^{CreERT2/+};Sox17^{fl/fl}*) adult mutant mice though (*Figure 3—figure supplement 2*), suggesting that the phenotype in *Pax3^{Cre/+};Sox17^{GFP/fl}* mice was linked to impaired early postnatal growth and satellite cell-derived myonuclear accretion (*White et al., 2010*). These results demonstrate that SOX17 plays an important role in induction and maintenance of satellite cell quiescence.

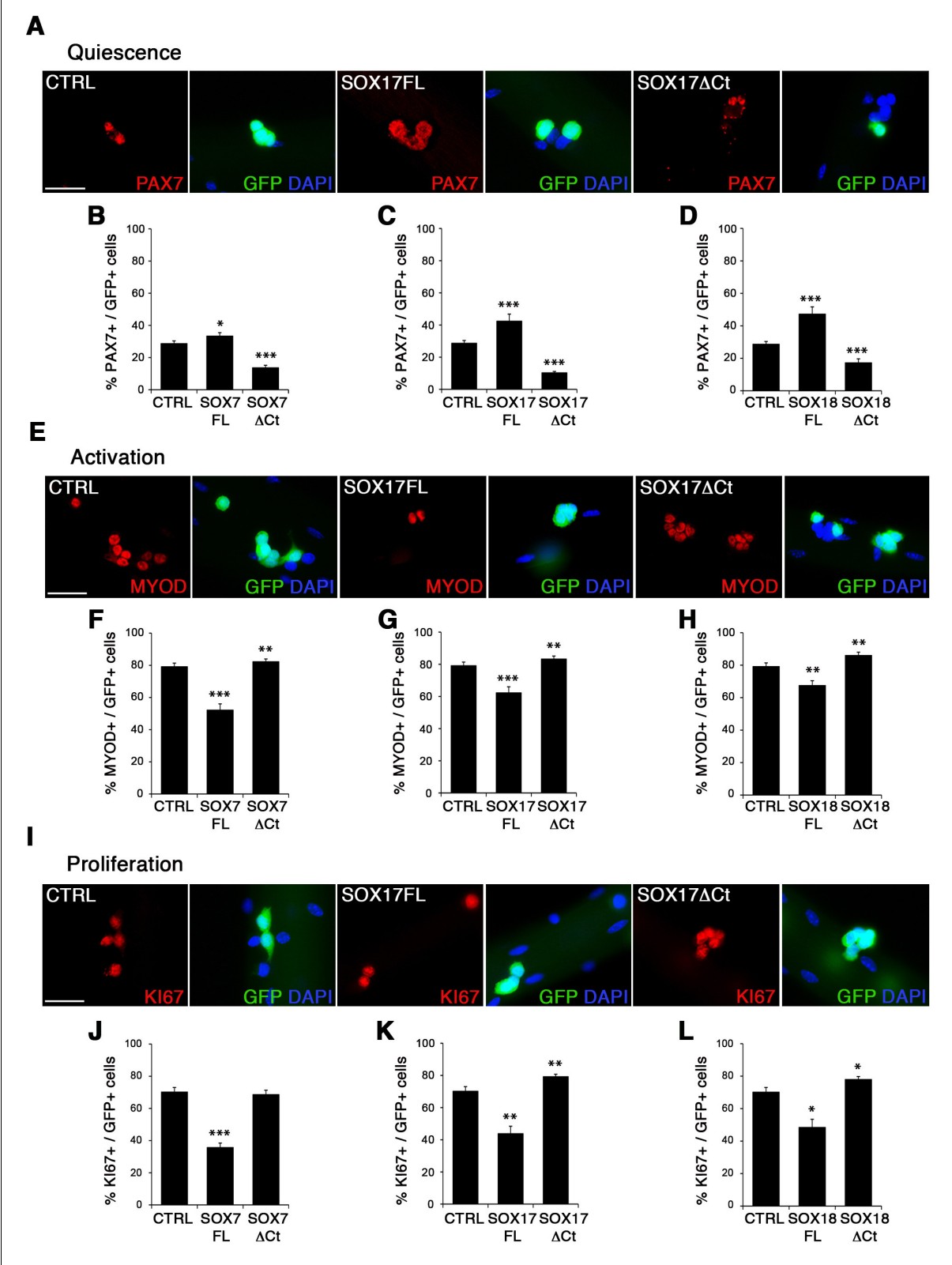

**Figure 2.** SOXF factors modulate satellite cell behavior. (A–E–I) Immunofluorescence of satellite cells transduced with SOXF-encoding retroviruses after 72 hr in culture on isolated adult wild type EDL myofibers. SOXF-FL, construct overexpressing SOXF; SOXFΔCt, altered construct lacking the C-terminus (preserving the HMG DNA binding domain); CTRL, encoding just eGFP. GFP marks transduced cells. Nuclei are counterstained with DAPI (blue). Scale bars, 20 μm. (B–D, F–H, J–L) Quantification of the transduced satellite cells illustrated in (A–E–I) for quiescence (PAX7), activation (MYOD),

*Figure 2 continued on next page*

*Figure 2 continued*

and proliferation (KI67), compared to CTRL. n ≥ 50 fibers/EDL per condition; ≥1000 satellite cells/EDL. Data expressed as mean ± s.e.m., statistically analyzed with Student's unpaired t-test: *, p<0.05; **, p<0.01; ***, p<0.001, compared to CTRL.

DOI: https://doi.org/10.7554/eLife.26039.004

The following figure supplement is available for figure 2:

**Figure supplement 1.** SoxF gene function in satellite cell homeostasis.

DOI: https://doi.org/10.7554/eLife.26039.005

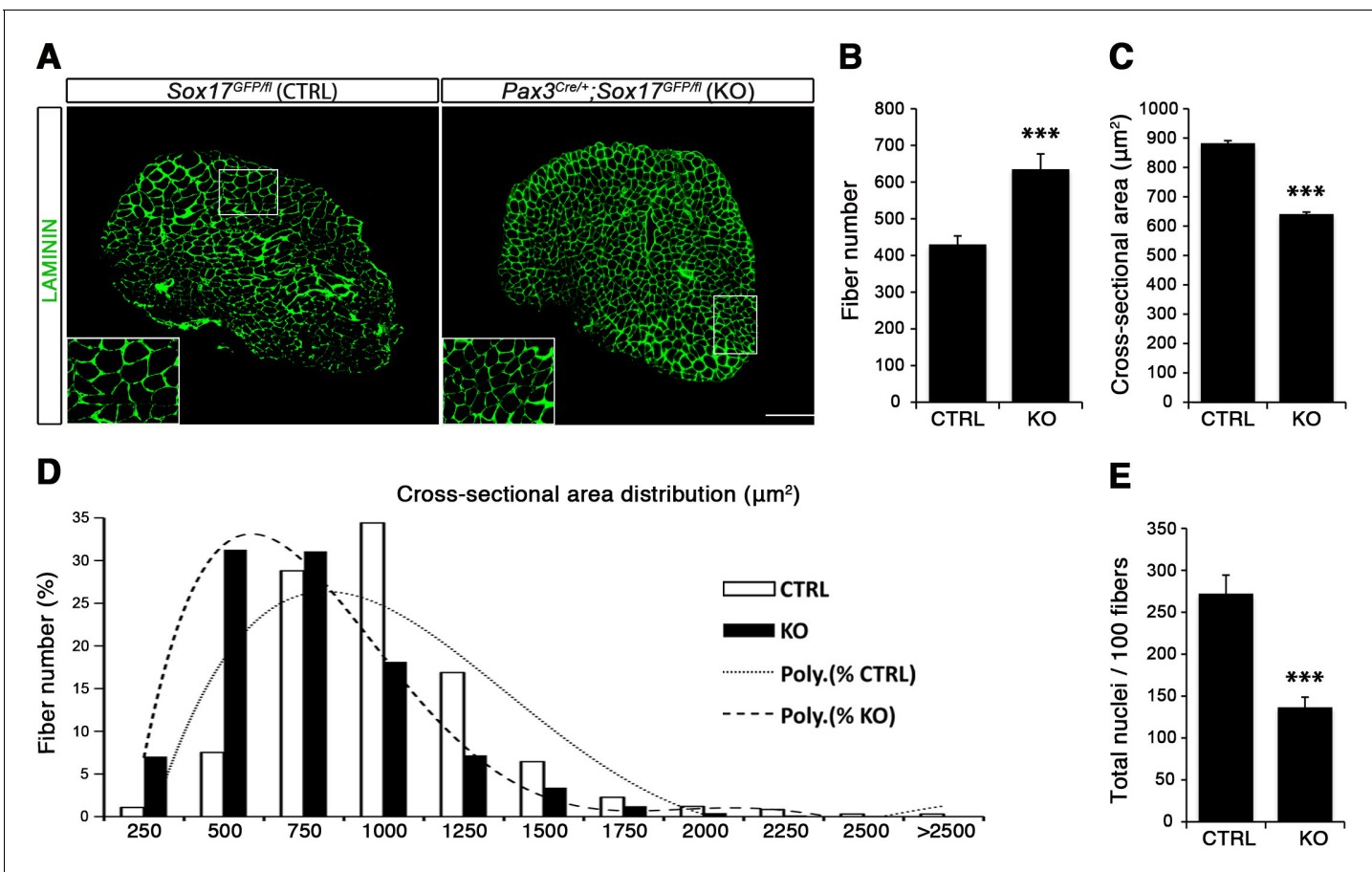

**Figure 3.** *Sox17*-knockout during prenatal establishment of satellite cells modifies adult myofiber content and morphology. (**A**) Representative *Soleus* muscle cryosection images of adult control and *Sox17* mutant mice. Immunofluorescence was performed with LAMININ to identify the myofibers. Higher magnification is shown in the boxed area. Scale bar, 200 μm. (**B–C**) Quantification of myofiber number (**B**) and cross-sectional area in μm² (**C**). (**D**) Distribution of the cross-sectional myofiber area in μm². 'Poly.', polynomial curve fitting the distribution of myofiber size. (**E**) Quantification of myonuclei number per 100 fibers in adult *Soleus* cross-sections from control and *Sox17*-knockout mice. CTRL, *Sox17^{GFP/fl}*; KO, *Pax3^{Cre/+};Sox17^{GFP/fl}*. n ≥ 4 mice (each quantified in triplicate) for all experiments. Data expressed as mean ± s.e.m., statistically analyzed with Student's unpaired t-test: ***, p<0.001, compared to CTRL.

DOI: https://doi.org/10.7554/eLife.26039.006

The following figure supplements are available for figure 3:

**Figure supplement 1.** Muscle characterization in control and *Sox17*-knockout mice.

DOI: https://doi.org/10.7554/eLife.26039.007

**Figure supplement 2.** Muscle characterization in control and *Sox17*-conditional knockout mice.

DOI: https://doi.org/10.7554/eLife.26039.008

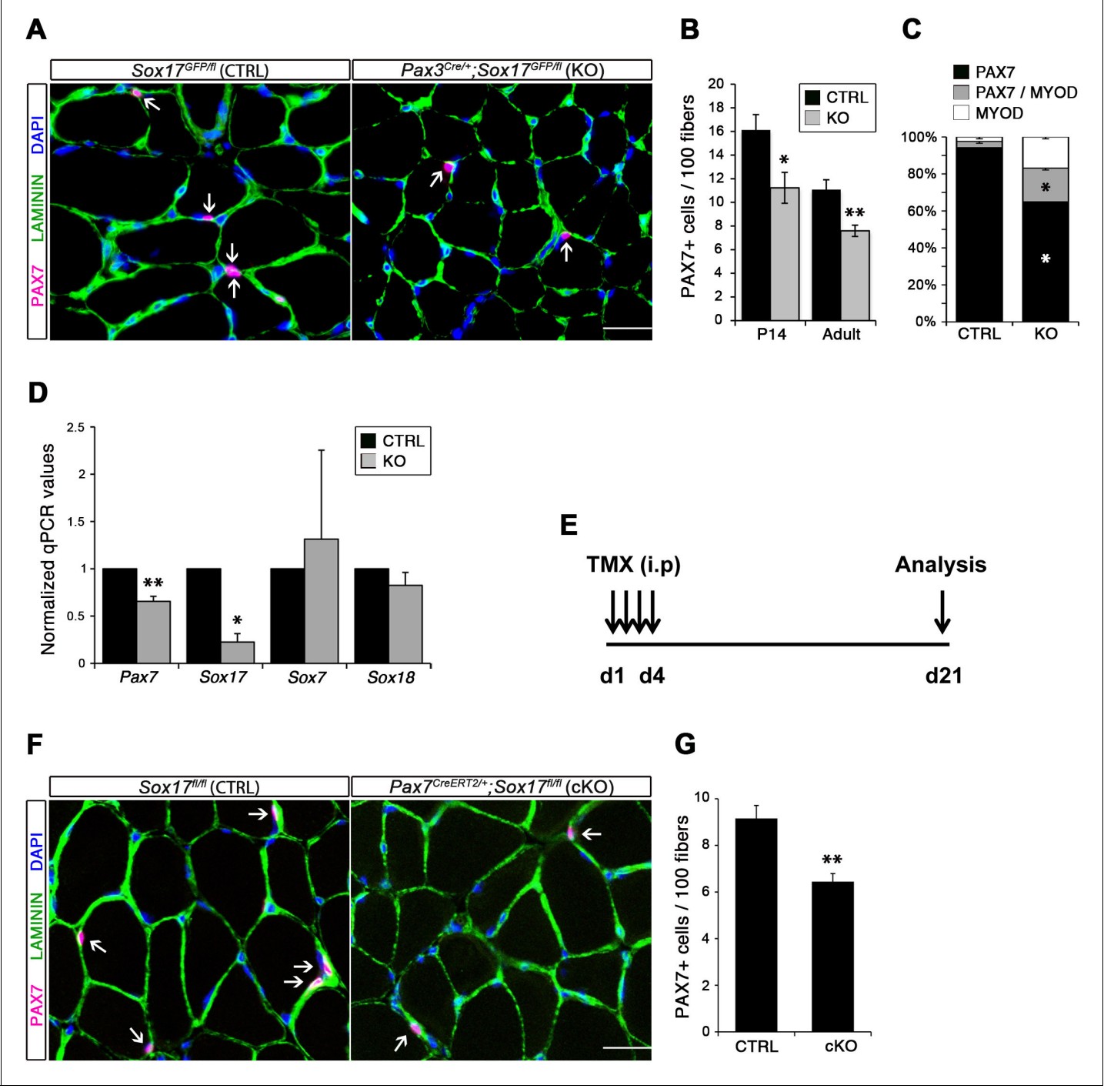

**Figure 4.** SOX17 is necessary to maintain satellite cell quiescence in adult muscles. (A,F) Representative *Soleus* cryosection images showing immunofluorescence for satellite cells (PAX7+, arrows) in *Pax3^Cre/+;Sox17^GFP/fl* and *Pax7^CreERT2/+;Sox17^fl/fl* mice, with appropriate controls. Scale bars, 25 μm. Fibers are identified by LAMININ and nuclei are counterstained with DAPI. (B,G) Quantification of satellite cell number during postnatal growth (P14) and in adult. (C) Quantification of the ratio PAX7/MYOD+ satellite cells in P14 *Soleus* cryosections. (D) RT-qPCR analysis on adult TA muscles for *Pax7* and SoxF genes in fresh FACS-isolated satellite cells from control and *Sox17*-knockout mice. (A–D) CTRL, *Sox17^GFP/fl*; KO, *Pax3^Cre/+;Sox17^GFP/fl*. (E) Schematic outline of the experimental procedure for tamoxifen (TMX) injection (i.p., intraperitoneal) in *Sox17^fl/fl* (CTRL) and *Pax7^CreERT2/+;Sox17^fl/fl* (cKO) mice. d, days. (E–G) CTRL, *Sox17^fl/fl*; cKO, *Pax7^CreERT2/+;Sox17^fl/fl*. Quantification was performed in whole cross-sections. n ≥ 4 mice (each quantified in triplicate) for all experiments. Data expressed as mean ± s.e.m., statistically analyzed with Student's unpaired t-test: *, p<0.05; **, p<0.01, compared to CTRL.

DOI: https://doi.org/10.7554/eLife.26039.009

*Figure 4 continued on next page*

*Figure 4 continued*

The following figure supplement is available for figure 4:

**Figure supplement 1.** Satellite cells characterization of control and *Sox17*-knockout mice.

DOI: https://doi.org/10.7554/eLife.26039.010

## Myogenic stem cell function is impaired during muscle regeneration in *Sox17*-deficient mice

To evaluate the role of SOX17 during satellite cell activation, renewal and differentiation in vivo, we carried out skeletal muscle regeneration assays. Following cardiotoxin (CTX)-induced regeneration in *Tibialis anterior* (TA) muscle of wild type mice, we first assessed the dynamics of SoxF gene expression by RT-qPCR in total injured muscle. We observed progressive up-regulation of SoxF genes, with distinct peaks at days (d) 4, 6, and 15 following injury (*Figure 5—figure supplement 1A*). Noticeably, d4 and d6 expression peaks coincided with increased levels of satellite cell markers such as *Pax7* and *Myf5* (*Figure 5—figure supplement 1B*), and at d4 with the myogenic regulatory factors *Myod* and *Myog* (*Figure 5—figure supplement 1C*), which mark activated satellite cells in the process of proliferation and differentiation to form new myofibers. Specific isolation of satellite cells using *Tg:Pax7-nGFP* (*Rocheteau et al., 2012*) through muscle regeneration depicts an identical behavior of all SoxF transcripts, being downregulated upon injury, and induced as regeneration proceeds (*Figure 5A*). SoxF genes and *Pax7* display a similar profile, contrary to commitment and differentiation markers (*Myod* and *Myog*, *Figure 5—figure supplement 1D*), inferring that SOXF have stem cell specific activity during regenerative myogenesis (*Figure 5A*).

Regenerating TA muscles in *Pax3^{Cre/+};Sox17^{GFP/fl}* mice were strikingly smaller than controls and expressed lower levels of myogenic genes (*Figure 5B–C*). Furthermore, we observed a loss of quiescence in *Sox17*-knockout satellite cells after muscle regeneration, likely preventing cells from re-establishing the pool of quiescent satellite cells (*Figure 5D–E*) so that when regeneration was over, the satellite cell pool was smaller in *Sox17*-knockout mutants (*Figure 5F*). Interestingly, when plating fresh FACS-sorted isolated satellite cells *in vitro*, *Sox17*-knockout cells proliferated more than control cells, yielding bigger colonies (*Figure 5—figure supplement 2A–B*). This result mimicked the effect obtained in satellite cells transduced with SOXFΔCt, with increased satellite cell proliferation at the expense of self-renewal (*Figure 2*). Histological analysis of TA muscles in *Pax3^{Cre/+};Sox17^{GFP/fl}* mice at d7 after CTX-induced regeneration revealed cell infiltration, fat accumulation and fibrosis, that were absent in regenerating muscles of control *Sox17^{GFP/fl}* mice (*Figure 5G*), suggesting abnormal regeneration and impaired satellite cell function (*Mann et al., 2011*; *Sambasivan et al., 2011*). Moreover, this delay in regeneration was still observed at d28, with signs of cell infiltration still evident (*Figure 5—figure supplement 2C–D*). However, a second injury at d28 did not exacerbate the phenotype seven days later (*Figure 5—figure supplement 2C,E*).

To confirm that muscle regeneration defect in *Pax3^{Cre/+};Sox17^{GFP/fl}* mice was due to satellite cell function compromised by loss of SOX17, we also examined regeneration in TA muscles of *Pax7-CreERT2/+;Sox17^{fl/fl}* mice (*Figure 6A*). Analysis of regeneration at d7 in *Sox17*-conditional knockout mutants revealed that satellite cell numbers were reduced, with fewer in quiescence (*Figure 6B–D*). At d28, diminution of the satellite cell pool was confirmed in regenerating muscle of adult conditional *Pax7^{CreERT2/+};Sox17^{fl/fl}* mutant mice (*Figure 6E–G*) as observed with *Pax3^{Cre/+};Sox17^{GFP/fl}* mice. Again, consistent with the phenotype in *Pax3^{Cre/+};Sox17^{GFP/fl}* mice, histological analysis of regenerated *Sox17*-conditional knockout TA muscles revealed cell infiltration, fat and fibrosis deposition, that were absent in regenerating muscles of control *Sox17^{fl/fl}* mice (*Figure 6H–L*), confirming abnormal regeneration and impaired satellite cell function in the absence of SOX17.

## Impaired SOXF function leads to severe muscle regeneration defects

Both myofiber culture and in vivo experiments suggested that SOXF factors are involved in satellite cell self-renewal. Alterations of SoxF gene function in myofiber culture experiments yielded stronger phenotypes than in vivo genetic ablation of just *Sox17*, suggesting a compensatory mechanism between SOX17 and other SOXF proteins. To study such a possible compensatory effect between SOXF members, we performed myofiber culture experiments in control *Sox17^{GFP/fl}* and *Pax3^{Cre/+};Sox17^{GFP/fl}* mutant mice, and analyzed the effect of expressing each of the SOXF factors

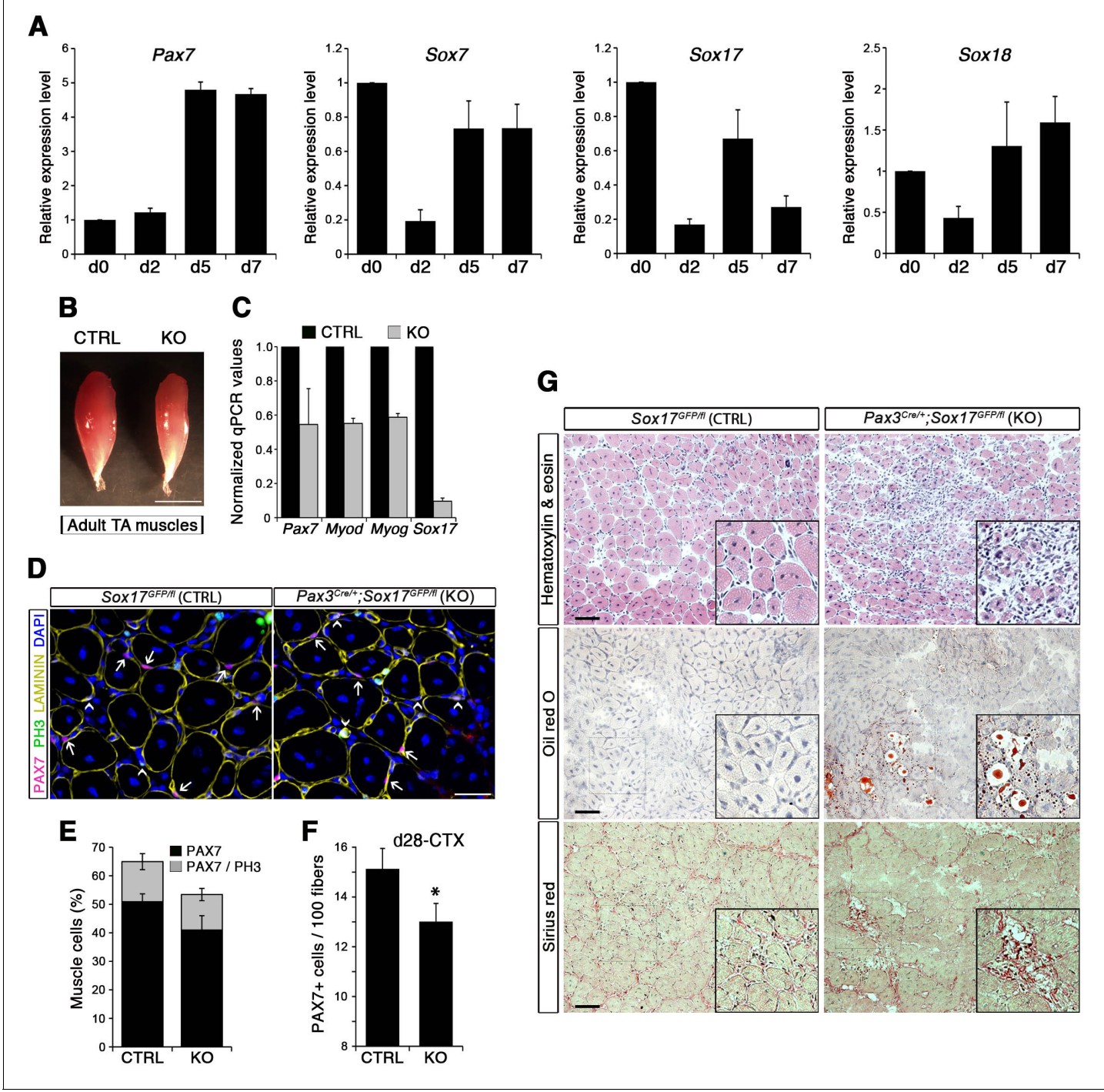

**Figure 5.** SOX17 regulates adult muscle regeneration after injury in *Pax3^Cre/+^;Sox17^GFP/fl* mutant mice. (**A**) RT-qPCR analysis of *Pax7* and SoxF genes in satellite cells isolated during CTX-induced regeneration in adult wild type TA muscles. d; days post-injury. (**B**) Representative images of TA muscles 10 days after CTX injection. Scale bar, 5 mm. (**C**) RT-qPCR of muscle markers 10 days after CTX injection. (**D**) Representative images of cryosections from regenerating adult TA muscles seven days after injury showing immunofluorescence for PAX7+ cells (quiescent; arrows) and PH3+PAX7+ cells (proliferating, arrowheads). Scale bar, 25 μm. (**E**) Quantification of satellite cells as illustrated in (**D**). (**F**) Quantification of satellite cells (PAX7+) by the end of the regeneration process (d28-CTX). (**G**) Representative images of the histological characterization of adult TA muscles seven days after injury with Hematoxylin and eosin (cell infiltration; upper panel), Oil red O (fat infiltration; middle panel), and Sirius red (fibrosis; bottom panel) staining. Insets: enlargement of the indicated regions. Scale bars, 100 μm. CTRL, *Sox17^GFP/fl*; KO, *Pax3^Cre/+^;Sox17^GFP/fl*. n ≥ 3 mice (each quantified in triplicate) for all experiments. Data expressed as mean ± s.e.m., statistically analyzed with Student's unpaired t-test: *, p<0.05, compared to CTRL.

DOI: https://doi.org/10.7554/eLife.26039.011

*Figure 5 continued on next page*

*Figure 5 continued*

The following figure supplements are available for figure 5:

**Figure supplement 1.** Gene expression profile during CTX-induced regeneration in adult wild type TA muscles and satellite cells.

DOI: https://doi.org/10.7554/eLife.26039.012

**Figure supplement 2.** Impaired clonogenic and regenerative potential of *Sox17*-knockout muscle stem cells.

DOI: https://doi.org/10.7554/eLife.26039.013

(*Figure 7A–C*). Consistent with the data shown in *Figure 4*, *Sox17* mutant satellite cells displayed reduced self-renewal (PAX7+/GFP+) (*Figure 7A*, CTRL vs. KO), associated with increased activation (MYOD+/GFP+) (*Figure 7B*, CTRL vs. KO), and little effect on differentiation (MYOG+/GFP+) (*Figure 7C*, CTRL vs. KO). Interestingly, transduction with retrovirus encoding either SOX7 or SOX17 rescued this defect in self-renewal, whereas expression of SOX18 was unable to revert this effect (*Figure 7A*). Moreover, overexpression of SOX7 or SOX17 strongly decreased the number of activated satellite cells, to even lower levels compared to control animals (*Figure 7B*). Expression of SOX18, however, did not modify the activation status of the cells. Finally, overexpression of each SOXF proteins induced a strong decrease in differentiation (*Figure 7C*), as previously observed in wild type cells (*Figure 2—figure supplement 1C–F*). These results demonstrate that overexpression of SOX7 or SOX17, but not SOX18, rescues the quiescence and activation phenotype of *Sox17*-knockout satellite cells.

To further characterize the redundant activity of SoxF genes in vivo, we took advantage of the dominant negative effect of SOX17ΔCt (*Figure 2*) to carry out electroporation into regenerating muscle (*Figure 7D–F*). Two days after CTX injection of wild type TA muscles, we electroporated a bi-cistronic construct co-expressing SOX17ΔCt and GFP (*Figure 7D* and *Figure 7—figure supplement 1*), together with a *TdTomato* reporter that revealed efficient electroporation along the regenerating muscle (*Figure 7—figure supplement 1*). Post-electroporation, we observed many areas of regenerating muscle devoid of fibers, with accumulation of fat and fibrosis, compared to control, indicating a general failure of muscles to regenerate (*Figure 7E*). A dramatic reduction in *Pax7* expression was associated with the exacerbated phenotype of SOX17ΔCt electroporated into muscle, compared to regeneration in *Sox17*-knockouts (*Figures 5C,G and* and *7E–F*). These results are consistent with SOXF activity being required for skeletal muscle regeneration and confirm the overlapping role of SOXF members, as previously reported in other tissues (*Matsui et al., 2006*; *Sakamoto et al., 2007*; *Sarkar and Hochedlinger, 2013*).

## Inhibition of β-catenin activity by SOXF factors in muscle stem cells

SOXF and β-catenin (CTNNB1) interact through a site located in the C-terminus of SOXF proteins (*Figure 2—figure supplement 1A*) and that deletion of this region is sufficient to ablate SOXF - β-catenin interaction (*Guo et al., 2008*; *Sinner et al., 2007*; *Sinner et al., 2004*; *Zhang et al., 2005*). Moreover, expression of constitutively active β-catenin in satellite cells in vivo leads to reduced myofiber size (*Hutcheson et al., 2009*; *Kuroda et al., 2013*), a phenotype similar to that we observe with the ablation of SOX17 in these cells (*Figure 3*). This suggests that SOXF inhibition of β-catenin activity could be required for muscle homeostasis. Upon activation of Wnt signaling, non-phosphorylated β-catenin is stabilized and translocates to the nucleus where it associates with TCF/LEF transcription factors to regulate target gene expression (*MacDonald et al., 2009*).

We designed two transcriptional reporter assays in C2C12 myoblasts to further characterize the SOXF - β-catenin interaction following β-catenin canonical signaling activation by LiCl (*Figure 8A–B*). All SOXF proteins individually, strongly activated our novel SoxF reporter, *SoxF-B-TKnLacZ* (containing five multimerized SOXF consensus binding motifs), demonstrating binding to the same consensus sequence (*Figure 8A*). Upon β-catenin co-expression with SOXF proteins, *SoxF-B-TKnLacZ* transactivation was further increased (*Figure 8A*). Conversely, we explored the role of SOXF proteins on LEF/TCF-β-catenin transcriptional activity (*Figure 8B*). In this system, β-catenin expression led to a four-fold increase in β-catenin reporter *pTOP-TKnLacZ* activity, while co-expression of SOXF impaired β-catenin-mediated induction of this reporter (*Figure 8B*). These functional assays indicate that while β-catenin enhances the transactivation activity of SOXF members, SOXF proteins hinder β-catenin-mediated activation of a TCF/LEF reporter in myogenic cells. Hence, our results imply that

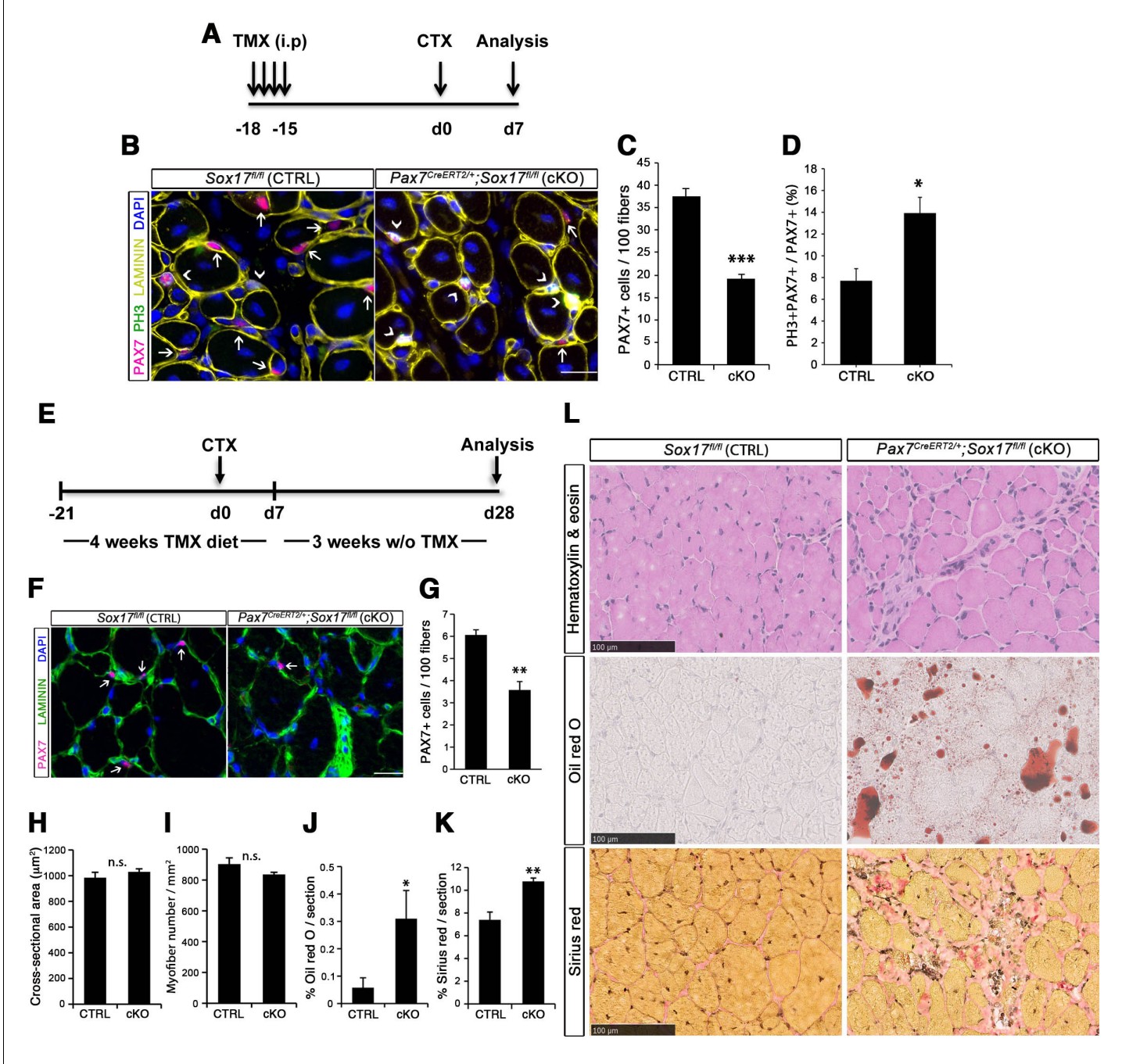

**Figure 6.** SOX17 regulates adult muscle regeneration after injury in *Pax7^CreERT2/+;Sox17^fl/fl* mutant mice. (**A**) Schematic outline of the experimental procedure for tamoxifen (TMX) injection (i.p., intraperitoneal). CTX, cardiotoxin injection; d, days. (**B**) Representative images of cryosections from regenerating adult TA muscles d7 after injury, showing immunofluorescence for PAX7+ (quiescent, arrows) and PH3+PAX7+ (proliferating, arrowheads) cells. Scale bar, 25 μm. (**C–D**) Quantification of satellite cells as illustrated in (**B**). (**E**) Schematic outline of the experimental procedure for TMX diet. CTX, cardiotoxin injection; d, days. (**F**) Representative images of cryosections from regenerating adult TA muscles d28 after injury, showing immunofluorescence for PAX7+ (quiescent, arrows) cells. Scale bar, 25 μm. (**G**) Quantification of satellite cells as illustrated in (**F**). (**H–I**) Quantification of the cross-sectional area in μm² (**H**) and myofiber number per mm² (**I**). (**J–K**) Quantification of fat infiltration (Oil red O) (**J**) and fibrosis (Sirius red) (**K**) indicated as proportion of the stained section (average of five sections per muscle). (**L**) Representative images of the histological characterization of adult TA muscles 28 days after injury with Hematoxylin and eosin (cell infiltration; upper panel), Oil red O (fat infiltration; middle panel), and Sirius red (fibrosis; bottom panel) staining. Scale bars, 100 μm. CTRL, *Sox17^fl/fl*; cKO, *Pax7^CreERT2/+;Sox17^fl/fl*. n ≥ 3 mice (each quantified at least in triplicate) for all experiments. Data expressed as mean ± s.e.m., statistically analyzed with Student's unpaired t-test (**C,D,G**) and Mann-Whitney ranking test (**H–K**): n. s., not significant; *, p<0.05; **, p<0.01; ***, p<0.001, compared to CTRL.

DOI: https://doi.org/10.7554/eLife.26039.014

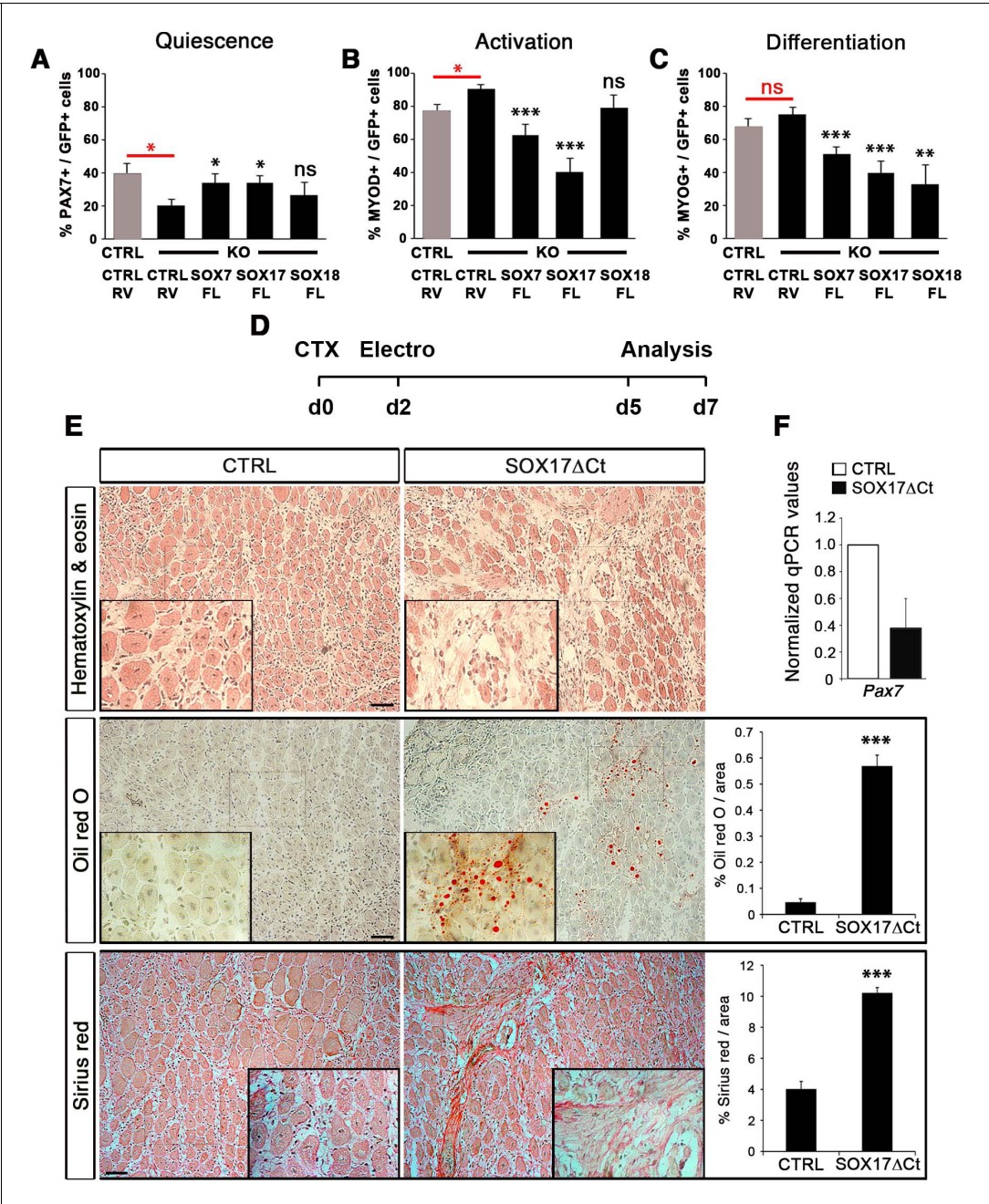

**Figure 7.** Compensatory effect of SOXF factors in satellite cells on *ex vivo* culture and in vivo injury-induced regeneration. (A–C) Quantification of transduced satellite cells with SOXF-encoding retroviruses after 72 hr in culture on EDL isolated myofibers. Adult control satellite cells were transduced with the eGFP-encoding retrovirus (CTRL-RV) and *Sox17*-knockout cells with CTRL-RV or SOXF-FL. Quiescence (A; PAX7), activation (B; MYOD), and differentiation (C; MYOG) were measured. In red, CTRL vs. KO comparison; in black, KO transduced with CTRL-RV vs. KO transduced with SOXF-FL. n ≥ 30 fibers/EDL per condition; ≥1000 satellite cells/EDL. CTRL, *Sox17*$^{GFP/fl}$; KO, *Pax3*$^{Cre/+}$;*Sox17*$^{GFP/fl}$. (D) Schematic outline of the experimental procedure for electroporation into regenerating TA muscle of wild type mice. CTX, cardiotoxin; d, days. (E) Histology characterization by Hematoxylin and eosin (cell infiltration, top panel), Oil red O (fat infiltration, middle panel), and Sirius red (fibrosis, bottom panel) staining of cryosections from electroporated wild type adult TA muscles five days after injury. TA muscles were electroporated with control (CTRL, left) or dominant negative SOX17 construct (SOX17ΔCt, right). Insets show enlarged images of the indicated regions. Quantification of fat infiltration (Oil red O) and fibrosis (Sirius red) are indicated as proportion of stained area. Scale bars, 100 μm. (F) RT-qPCR analysis seven days after CTX injection. n ≥ 3 mice (≥ 5 different areas). Data expressed as mean ± s.e.m., statistically analyzed with Student's unpaired t-test: ns, not significant; *, p<0.05; **, p<0.01; ***, p<0.001, compared to CTRL-RV in CTRL (red asterisks in A-C), CTRL-RV in KO (black asterisks in A-C) or CTRL (E).

DOI: https://doi.org/10.7554/eLife.26039.015

*Figure 7 continued on next page*

*Figure 7 continued*

The following figure supplement is available for figure 7:

**Figure supplement 1.** Muscle electroporation during injury-induced muscle regeneration.

DOI: https://doi.org/10.7554/eLife.26039.016

SoxF genes modulate β-catenin signaling during myogenesis. Strikingly, expression levels of known target genes of the canonical β-catenin pathway appear modified in *Sox17*-knockout muscles (*Figure 8C*). Indeed, *Jun*, *Ccnd1*, and *Axin2* expression were all increased two- to ten-fold in *Sox17* mutant muscles (*Figure 8C*).

In agreement with previous reports (*Otto et al., 2008*; *Rudolf et al., 2016*), we observed nuclear β-catenin expression in activated, but not quiescent, satellite cells indicating that induction of canonical signaling is synchronous with the activation of satellite cells (*Figure 8D*). To assess the functional significance of β-catenin binding to SOXF proteins, retroviral constructs of SOXF lacking β-catenin binding domain (SOXFΔBCAT) were generated (*Figure 2—figure supplement 1A*). Expression of SOXFΔBCAT in wild type satellite cells *ex vivo* caused a significant decrease in self-renewal capacity and increased activation (*Figure 8E–J*). These results mirrored those obtained with SOXFΔCt (*Figure 2* and *Figure 2—figure supplement 1C–F*), demonstrating that this motif is required for normal muscle stem cell function. Importantly, transactivation ability of SOXFΔBCAT mutant constructs on SOXF target genes was retained, as shown using the *SoxF-B-TKnLacZ* reporter (*Figure 8—figure supplement 1A*), whereas β-catenin transactivation of *pTOP-TKnLacZ* was partially restored when compared to SOXF-FL constructs (*Figure 8—figure supplement 1B*). Thus, interaction between SOXF proteins and β-catenin regulates muscle stem cell behavior following activation.

## SOXF factors modulate β-catenin transcriptional activity in satellite cells

To further demonstrate the functional interplay between SOX17 and β-catenin transcriptional activity in myogenic stem cells, single myofiber-associated satellite cells were treated with LiCl. This induction of β-catenin signaling yielded an expansion of the activated satellite cell pool (CTRL, *Figure 9A*). Overexpression of *Sox17* (SOX17FL) abolished the expansion of satellite cells (*Figure 9A*), while SOX17ΔCt did not affect the enhanced LiCl-driven expansion. Similar results were obtained when using CHIR9902, a specific inhibitor of the Glycogen synthase kinase-3 (GSK3B), which targets β-catenin for degradation (*data not shown*) (*Ying et al., 2008*). Our findings point to modulation of cell cycle by SOXF activity: satellite cells fail to acquire quiescence when SOXF function is impaired in vivo and ex vivo. In accord with these observations, the cell cycle regulator *Ccnd1* (Cyclin-D1) was up-regulated in *Sox17*-knockout satellite cells but absent in wild type cells (*Figure 8C* and *Figure 9B*). We next investigated how SOXF proteins affect the β-catenin transcriptional regulation of two target genes found increased in *Sox17*-knockouts, *Ccnd1* [also a SOX17 target (*Lange et al., 2009*)] and *Axin2*. We designed a cell-based transcriptional reporter assay using either 1 kb of the 5'UTR of *Ccnd1* (*Ccnd1-nLacZ*), encompassing binding motifs for TCF/LEF and SOXF proteins, or 5.6 kb of the proximal *Axin2* promoter (*Axin2-nLacZ*) (*Figure 9C–D*). β-catenin expression increased activity of both *Ccnd1-nLacZ* and *Axin2-nLacZ* reporters following LiCl treatment, while co-expression of SOX17 impaired β-catenin-mediated induction of these two reporters in a dose-dependent manner (*Figure 9C–D*). SOX7ΔBCAT, lacking the β-catenin binding site, however, was unable to influence activation of either the *Ccnd1-nLacZ* or *Axin2-nLacZ* reporters. Accordingly, *Axin2* expression levels appeared to be progressively down-regulated at the onset of satellite cells emergence, thus displaying general inverse dynamics to SoxF genes (*Figure 9E*) (*Alonso-Martin et al., 2016*).

Together, our data demonstrate that SOXF factors control expansion and self-renewal of adult muscle stem cells, associated with an inhibition of TCF/LEF-β-catenin target genes.

## Discussion

We previously performed a global transcriptomic analysis of the changes in gene expression in murine muscle stem cells throughout life (*Alonso-Martin et al., 2016*). Focusing on the signature associated with establishment and maintenance of satellite cells from their developmental

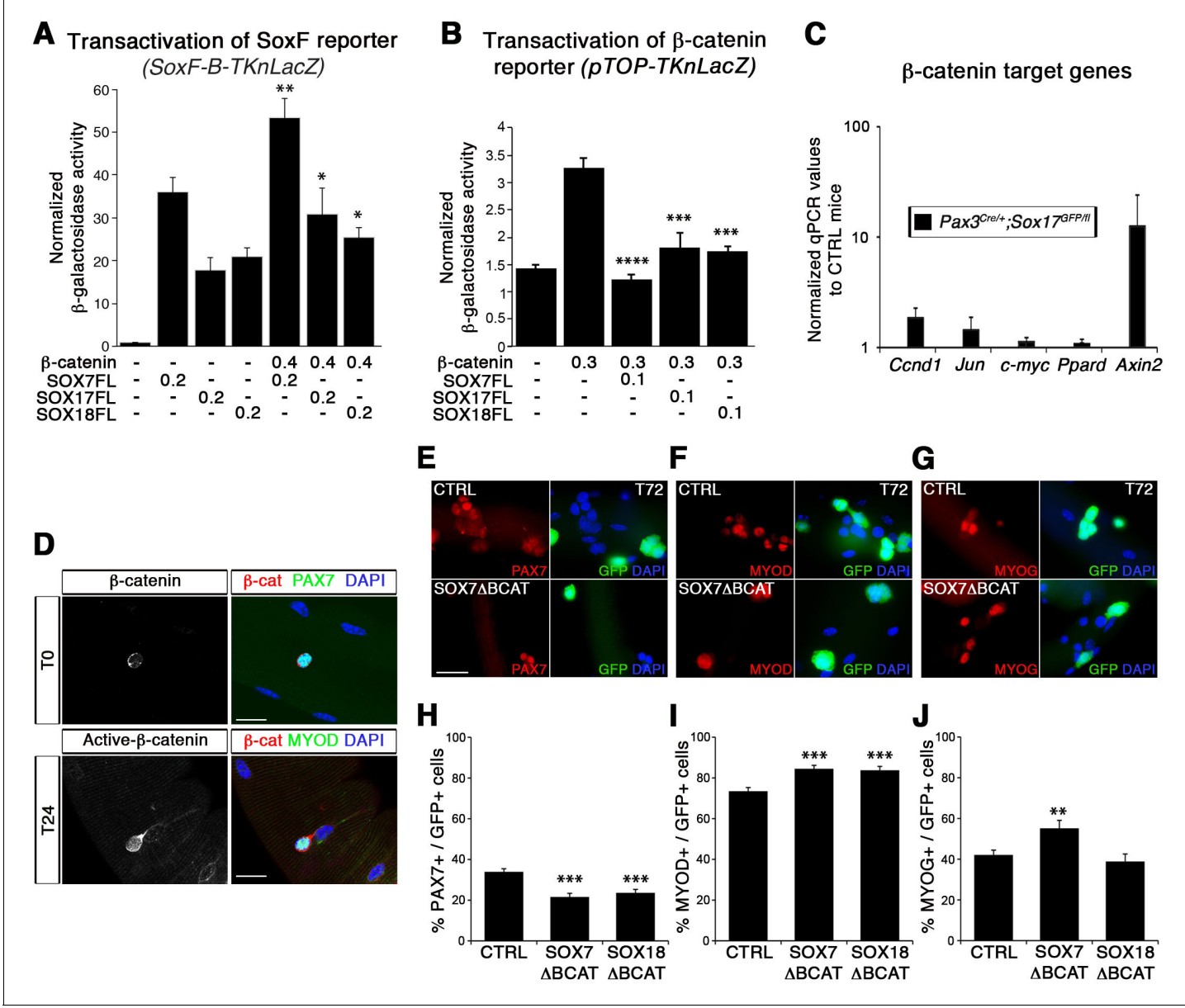

**Figure 8.** SoxF genes inhibit β-catenin transcriptional activity to regulate satellite cell behavior. (A–B) Transactivation of *SoxF-B-TKnLacZ* (A) and *pTOP-TKnLacZ* (B) reporters by SOXF and β-catenin in LiCl-treated C2C12 myoblasts. Quantification is expressed as mean of the amount (nmoles) of hydrolyzed ONPG normalized to control (first bar). Comparison of activity with or without β-catenin (A) or with and without SOXF co-expression (B). Relative amounts of transfected DNA are listed below the chart (ng). n ≥ 4 (A); n ≥ 6 (B). (C) Expression profile of β-catenin target genes in adult control and *Sox17* mutant TA muscles. *Ccnd1*, Cyclin-D1. n ≥ 4 mice (each in triplicate). (D) Immunolabeling for β-catenin (β-cat, red) in quiescent (T0, PAX7+, green) and activated (T24, MYOD+, green) satellite cells from adult wild type EDL isolated myofibers. Nuclei are counterstained with DAPI (blue). Scale bar, 50 µm. (E–G) Immunofluorescence of satellite cells transduced with SOXFΔBCAT constructs after 72 hr in culture (T72) in adult wild type EDL isolated myofibers. SOXFΔBCAT, SOXF-encoding retroviruses lacking the binding site for β-catenin; CTRL, encoding just eGFP. GFP indicates transduced cells. Nuclei are counterstained with DAPI (blue). Scale bars, 20 µm. (H–J) Quantification of the transduced satellite cells illustrated in (E–G) for quiescence (PAX7), activation (MYOD), and differentiation (MYOG; myogenin). n ≥ 50 fibers/EDL; ≥1000 satellite cells/EDL. Data expressed as mean ± s.e.m., statistically analyzed with Mann-Whitney ranking test (A–B) or Student's unpaired t-test (H-J): *, p<0.05; **, p<0.01; ***, p<0.001; ****, p<0.0001, compared to absence of β-catenin (A), presence of β-catenin (B) or CTRL retrovirus (H-J).

DOI: https://doi.org/10.7554/eLife.26039.017

The following figure supplement is available for figure 8:

**Figure supplement 1.** Validation of SOXF constructs.

DOI: https://doi.org/10.7554/eLife.26039.018

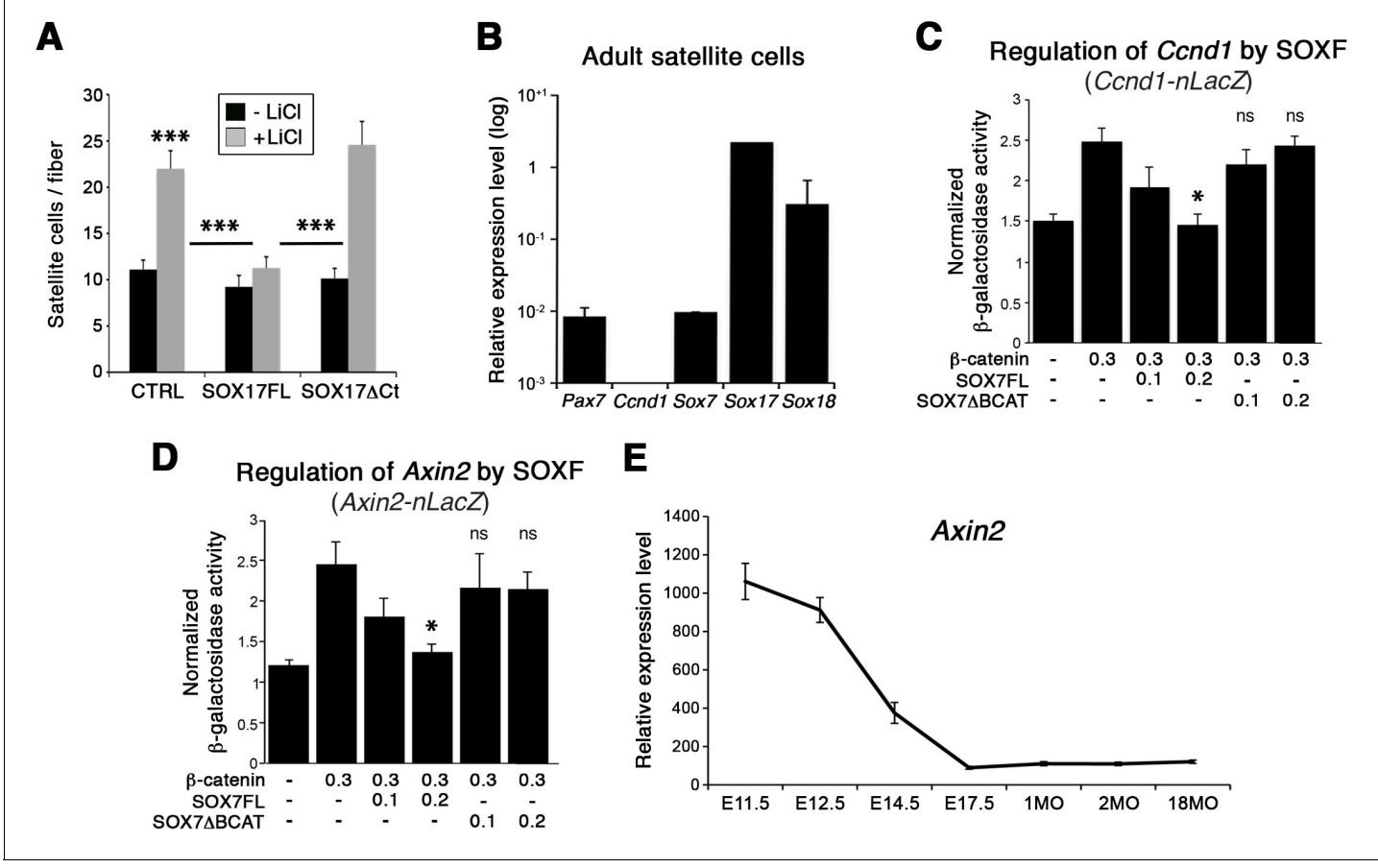

**Figure 9.** SOXF factors inhibit β-catenin target genes. (**A**) Effect of β-catenin stabilizer LiCl in adult wild type EDL myofiber cultures, to analyze satellite cell proliferation rate upon transduction with the indicated retroviral constructs. SOXF-FL, construct overexpressing SOXF; SOXFΔCt, SOXF proteins C-terminal deletions preserving the HMG DNA binding domain. n ≥ 50 fibers/EDL; ≥1000 satellite cells/EDL. (**B**) RT-qPCR of adult quiescent satellite cells. *Pax7* is the marker of this stem cell population. SoxF transcripts were detected but not *Ccnd1* (Cyclin-D1). n = 3. (**C–D**) Fold transactivation of *Ccnd1* (*Ccnd1-nLacZ*) (C; n = 3) or *Axin2* (*Axin2-nLacZ*) (D; n = 4) proximal promoters by β-catenin in C2C12 myoblasts co-transfected with SOX7 constructs in presence versus absence of LiCl. Quantification is expressed as mean of the amount (nmoles) of hydrolyzed ONPG normalized to control (first bar). Comparison is related to β-catenin only transfection. Relative amounts of transfected DNA are listed below the chart (ng). (**E**) Expression levels of *Axin2* in FACS-isolated *Pax3^{GFP/+}* cells from Affymetrix expression analysis. E, Embryonic day; P, Postnatal day; MO, age in months. Data expressed as mean ± s.e.m., statistically analyzed with Student's unpaired t-test (**A**) or Mann-Whitney ranking test (**C–D**): ns, not significant; *, p<0.05; ***, p<0.001, compared to absence of LiCl (A, CTRL), SOX17FL (A, LiCL treated CTRL and SOXFΔCt) or β-catenin only transfection (C-D).
DOI: https://doi.org/10.7554/eLife.26039.019

progenitors, we identified SoxF genes, *Sox7*, *Sox17*, and *Sox18* as of interest. SoxF transcripts become expressed at the time of satellite cell emergence, with a maximum expression in the quiescent adult state, highlighting their role in establishment, maintenance and function of muscle stem cells. Of relevance, SOX17 is involved in cell fate decisions in human primordial germ cells and embryo-derived stem cells (*Irie et al., 2015*; *McDonald et al., 2014*).

Absence of SOX17 leads to impaired postnatal muscle development, with an increase of smaller fibers. Postnatal muscle fiber hypertrophy depends on the total number of muscle fibers within a muscle; thus, the postnatal growth rate of the individual muscle fiber would be lower when there are more myofibers (*Rehfeldt et al., 2000*). In addition, the reduction of myonuclei per myofiber suggests that myofiber growth impairment may be due to a reduced contribution of satellite cell fusion (*White et al., 2010*). Consistent with these findings, we observed fewer satellite cells in *Sox17*-knockout mice, associated with a loss of quiescence and a reduced stem cell pool in postnatal muscles. Moreover, when SOXF function is impaired in satellite cells, self-renewal capacity is reduced and both activation and proliferation are increased. Satellite cell self-renewal is critical to maintain

the pool of the satellite cells, so impairment of this process translates into reduced cell numbers, resulting in defective muscle regeneration in both $Pax3^{Cre/+};Sox17^{GFP/fl}$ and $Pax7^{CreERT2/+};Sox17^{fl/fl}$ mutant mice, highlighting the specific relevance of SoxF genes postnatally and specifically in adult satellite cells. Moreover, we show that SOXF overexpression in satellite cells inhibits proliferation and differentiation and promotes self-renewal, with SOX17 promoting self-renewal in other stem cell types, such as adult hematopoietic progenitors (*Chhabra and Mikkola, 2011*; *He et al., 2011*).

Specific genetic ablation of *Sox17* leads to milder phenotypes than when dominant negative constructs are used, which suppress transcriptional activation through all SOXF proteins, in myofiber cultures (*ex vivo*) or injured muscle electroporation (in vivo). Yet, despite apparently normal expression of *Sox7* and *Sox18* in *Sox17* mutant mice (*Figure 4D*), there is a general loss of quiescence in satellite cells. SoxF genes have been reported to act with redundant functions, as versatile regulators of embryonic development and determination of different stem and progenitor cell fate (*Matsui et al., 2006*; *Sakamoto et al., 2007*; *Sarkar and Hochedlinger, 2013*). However, our data suggest that in muscle stem cells, redundancy between SoxF genes is more complex. For instance, overexpression of SOX7 or SOX17 but not SOX18 is sufficient to rescue the phenotype in *Sox17* mutant mice. Recently, a $Sox7^{fl}$ mutant mouse has been reported, revealing the genetic interaction of SOX7 with SOX17 in developmental angiogenesis (*Kim et al., 2016*). Furthermore, during revisions for this study, a muscle-specific ablation of *Sox7* ($Pax3^{Cre/+};Sox7^{fl/fl}$) was reported, showing upregulation of *Sox17* and *Sox18* in the absence of *Sox7* (*Rajgara et al., 2017*). Nevertheless, *Sox7*-deficient muscles demonstrated severe phenotypes in homeostatic and regeneration conditions (*Rajgara et al., 2017*), similar to *Sox17* ablation in myogenic cells (*Figures 3–6*). Future studies analyzing the impact of ablating both SOX7 and SOX17 for muscle stem cell function will be of interest.

Finally, our data link SOXF regulation of satellite cell self-renewal with control of β-catenin activity in satellite cells. Interaction between SOXF and β-catenin has been reported in other cell types, i.e. repression of β-catenin-stimulated expression of dorsal genes (*Zorn et al., 1999*), regulation of endodermal genes (*Sinner et al., 2004*), or acting as tumor suppressors antagonizing Wnt/β-catenin signaling (*Liu et al., 2016*; *Sinner et al., 2007*; *Takash et al., 2001*), as well as regulators of this pathway in oligodendrocyte progenitor cells (*Chew et al., 2011*; *Ming et al., 2013*). More importantly, our data provide a molecular mechanism for previous reports which demonstrate that a tight regulation of the Wnt/β-catenin canonical signaling output is required to ensure skeletal muscle regeneration (*Brack et al., 2008*; *Brack et al., 2007*; *Figeac and Zammit, 2015*; *Murphy et al., 2014*; *Otto et al., 2008*; *Parisi et al., 2015*; *Rudolf et al., 2016*; *Seale et al., 2003*; *von Maltzahn et al., 2012*). Hence, SOXF factors display a dual activity as both intrinsic regulators of muscle stem cell quiescence and interacting with extrinsic signaling pathways to regulate the expansion of activated muscle stem cells. Moreover, recent findings demonstrate that old satellite cells are incapable of maintaining their normal quiescent state in muscle homeostatic conditions, by switching to an irreversible pre-senescence state (*Sousa-Victor et al., 2014*). Satellite cells fail to regulate their quiescence with aging, leading to depletion of the pool of stem cells (*Blau et al., 2015*). Interestingly, satellite cell functional impairment is associated with up-regulation of canonical Wnt/β-catenin (*Brack et al., 2008*; *Brack et al., 2007*). Our data therefore points to a potential role of SOXF-β-catenin interaction in this context.

In conclusion, we demonstrate that SOXF transcription factors play a key role in stem cell quiescence and myogenesis through both direct transcriptional control and by modulation of the output of β-catenin activity to affect canonical Wnt signaling.

## Materials and methods

### Key resources table

| Reagent type (species) or resource | Designation | Source or reference | Identifiers | Additional information |
|---|---|---|---|---|
| Gene (*Mus musculus*) | *Sox7* | I.M.A.G.E. clone | 40131228 | N/A |

*Continued on next page*

Continued

| Reagent type (species) or resource | Designation | Source or reference | Identifiers | Additional information |
|---|---|---|---|---|
| Gene (*Mus musculus*) | *Sox18* | I.M.A.G.E. clone | 3967084 | N/A |
| Strain, strain background (*Mus musculus*) | *Pax3^GFP/+^* | PMID: 15843801 DOI: 10.1038/nature03594 | N/A | Mouse line maintained in F. Relaix lab |
| Strain, strain background (*Mus musculus*) | *Pax3^Cre/+^* | The Jackson Laboratory PMID: 15882581 DOI: 10.1016/j.ydbio.2005.02.002 | *B6;129-Pax3^tm1(cre)Joe^/J* MGI: J:96431 RRID:IMSR_JAX:005549 | Mouse line obtained from J. A. Epstein |
| Strain, strain background (*Mus musculus*) | *Pax7^CreERT2/+^ (Pax7^+/CE^)* | The Jackson Laboratory PMID: 19554048 PMCID: PMC2767162 DOI: 10.1038/nature08209 | *B6;129-Pax7^tm2.1(cre/ERT2)Fan^/J* MGI: J:150962 RRID:IMSR_JAX:012476 | Mouse line obtained from C.M. Fan |
| Strain, strain background (*Mus musculus*) | *Tg:Pax7-nGFP* | PMID: 22265406 DOI: 10.1016/j.cell.2011.11.049 | *Tg(Pax7-EGFP)#Tajb* MGI:5308730 RRID:MGI:5308742 | Mouse line obtained from S. Tajbakhsh |
| Strain, strain background (*Mus musculus*) | *Sox17^GFP/+^* | The Jackson Laboratory PMID: 17655922 PMCID: PMC2577201 DOI: 10.1016/j.cell.2007.06.011 | *BKa.Cg-Sox17^tm1Sjm^ Ptprc^b^ Thy1^a^/J* MGI: J:123050 RRID:IMSR_JAX:007687 | Mouse line obtained from S. J. Morrison |
| Strain, strain background (*Mus musculus*) | *Sox17^fl/+^* | The Jackson Laboratory PMID: 17655922 PMCID: PMC2577201 DOI: 10.1016/j.cell.2007.06.011 | *BKa.Cg-Sox17^tm2Sjm^ Ptprc^b^ Thy1^a^/J* MGI: J:123050 RRID:IMSR_JAX:007686 | Mouse line obtained from S. J. Morrison |
| Cell line (*Mus musculus*) | C2C12 | American Type Culture Collection (ATCC) PMID: 28966089 PMCID: PMC5640514 DOI: 10.1016/j.cub.2017.08.031 | CRL-1772 RRID: CVCL_0188 | Cell line maintained in E. Gomes lab |
| Antibody | anti-GFP (rabbit polyclonal) | Life Technologies | A11122 RRID:AB_221569 | 1:500 |
| Antibody | anti-GFP (chicken polyclonal) | Abcam | ab13970 RRID:AB_300798 | 1:500 |
| Antibody | anti-Ki67 (mouse monoclonal) | BD Pharmingen | 556003 RRID:AB_396287 | 1:100 |
| Antibody | anti-Ki67 (rabbit polyclonal) | Abcam | ab15580 RRID:AB_443209 | 1:100 |
| Antibody | anti-Laminin (rabbit polyclonal) | Sigma-Aldrich | L9393 RRID:AB_477163 | 1:100 |
| Antibody | anti-Laminin (AlexaFluor647) | Novus Biological | NB300-144AF647 | 1:200 |
| Antibody | anti-M-Cadherin (mouse monoclonal) | nanoTools | MCAD-12G4 | 1:50 |
| Antibody | anti-MyoD1 (5.8A) (mouse monoclonal) | DAKO | M3512 RRID:AB_2148874 | 1:50 |
| Antibody | anti-MyoD (M-318) (rabbit polyclonal) | Santa Cruz | sc-760 RRID:AB_2148870 | 1:20 |
| Antibody | anti-Myogenin (mouse monoclonal) | DSHB | F5D | 1:100 |
| Antibody | anti-Pax7 (mouse monoclonal) | DSHB | PAX7-c | 1:20 |
| Antibody | anti-Pax7 (mouse monoclonal) | Santa Cruz | sc-81648 RRID:AB_2159836 | 1:20 |
| Antibody | anti-Phospho-Histone H3 (Ser10) (rabbit polyclonal) | Merck Millipore | 06–570 RRID:AB_310177 | 1:500 |

*Continued*

| Reagent type (species) or resource | Designation | Source or reference | Identifiers | Additional information |
|---|---|---|---|---|
| Antibody | anti-Sox17 (goat polyclonal) | R and D Systems | AF1924 RRID:AB_355060 | 1:50 |
| Antibody | Alexa 488 goat anti-mouse IgG (H + L) | Life Technologies | A-11017; RRID:AB_143160 A-21121; RRID:AB_141514 | 1:400 |
| Antibody | Alexa 546 goat anti-mouse IgG (H + L) | Life Technologies | A-11018 RRID:AB_2534085 | 1:400 |
| Antibody | Alexa 555 goat anti-mouse IgG (H + L) | Life Technologies | A-21425 RRID:AB_2535846 | 1:400 |
| Antibody | Alexa 594 goat anti-mouse IgG (H + L) | Life Technologies | A-11020. RRID:AB_141974 A-21125; RRID:AB_141593 | 1:400 |
| Antibody | Alexa 488 goat anti-rabbit IgG (H + L) | Life Technologies | A-11070 RRID:AB_142134 | 1:400 |
| Antibody | Alexa 594 goat anti-rabbit IgG (H + L) | Life Technologies | A-11072 RRID:AB_142057 | 1:400 |
| Antibody | Alexa 594 donkey anti-goat IgG (H + L) | Life Technologies | A-11058 RRID:AB_142540 | 1:400 |
| Antibody | Alexa 488 goat anti-Chicken IgY (H + L) | Life Technologies | A-11039 RRID:AB_142924 | 1:400 |
| Antibody | Cy5-goat anti-rabbit IgG (H + L) | Jackson ImmunoResearch | 111-175-144 RRID:AB_2338013 | 1:200 |
| Antibody | Rat anti-mouse CD45-PE-Cy7 | BD Pharmingen | 561868 RRID:AB_10893599 | 10 ng/ml |
| Antibody | Rat anti-mouse Ter119-PE-Cy7 | BD Pharmingen | 557853 RRID:AB_396898 | 10 ng/ml |
| Antibody | Rat anti-mouse CD34-BV421 | BD Pharmingen | 562608 RRID:AB_11154576 | 10 ng/ml |
| Antibody | Rat anti-mouse integrin-$\alpha$7-A700 | R and D Systems | FAB3518N RRID:AB_10973483 | 10 ng/ml |
| Antibody | Rat anti-mouse Sca1-FITC | BD Pharmingen | 553335 RRID:AB_394791 | 10 ng/ml |
| Antibody | Rat anti-mouse CD31-PE | BD Pharmingen | 553373 RRID:AB_394819 | 10 ng/ml |
| Sequence-based reagent (Pax7_foward primer) | 5' – AGGCCTTCGAGAGG ACCCAC – 3' | Eurogentec | N/A | N/A |
| Sequence-based reagent (Pax7_reverse primer) | 5' – CTGAACCAGACCTG GACGCG – 3' | Eurogentec | N/A | N/A |
| Sequence-based reagent (Sox7_foward primer) | 5' – CTTCAGGGGACAA GAGTTCG – 3' | Eurogentec | N/A | N/A |
| Sequence-based reagent (Sox7_reverse primer) | 5' – GGGTCTCTTCTGG GACAGTG – 3' | Eurogentec | N/A | N/A |
| Sequence-based reagent (Sox17_foward primer) | 5' – GCCAAAGACGAACGC AAGCGGT – 3' | Eurogentec | N/A | N/A |
| Sequence-based reagent (Sox17_reverse primer) | 5' – TCATGCGCTTCACCT GCTTG – 3' | Eurogentec | N/A | N/A |

*Continued on next page*

Continued

| Reagent type (species) or resource | Designation | Source or reference | Identifiers | Additional information |
|---|---|---|---|---|
| Sequence-based reagent (Sox18_foward primer) | 5' – AACAAAATCCGGATC TGCAC – 3' | Eurogentec | N/A | N/A |
| Sequence-based reagent (Sox18_reverse primer) | 5' – CGGTACTTGTAGTTGGG ATGG – 3' | Eurogentec | N/A | N/A |
| Sequence-based reagent (Ccnd1_foward primer) | 5' – TTCCTCTCCTGCTA CCGCAC – 3' | Eurogentec | N/A | N/A |
| Sequence-based reagent (Ccnd1_reverse primer) | 5' – GACCAGCCTCTTCCTC CACTTC – 3' | Eurogentec | N/A | N/A |
| Sequence-based reagent (Axin2_fowardprimer) | 5' – AAGAGAAGCGACCCAGT CAA – 3' | Eurogentec | N/A | N/A |
| Sequence-based reagent (Axin2_reverse primer) | 5' – CTGCGATGCATCTCTC TCTG – 3' | Eurogentec | N/A | N/A |
| Sequence-based reagent (SoxF binding site) | 5' – CAACAATCATCATTGTTGG GGCCAACAATCTACATTGTT CAGA – 3' | Eurogentec | N/A | N/A |
| Sequence-based reagent (SoxF binding site) | 5' – TCTGAACAATGTAGATTGT TGGCCCCAACAATGATGATT GTTG – 3' | Eurogentec | N/A | N/A |
| Commercial assay or kit | LIVE/DEAD Fixable Blue Dead Cell Stain Kit | Life Technologies | L23105 | N/A |
| Commercial assay or kit | RNasy Micro Kit | QIAGEN | 74004 | N/A |
| Commercial assay or kit | RNeasy Fibrous Tissue Midi Kit | QIAGEN | 75742 | N/A |
| Commercial assay or kit | Transcriptor First Strand cDNA Synthesis Kit | Roche-Sigma-Aldrich | 04897030001 | N/A |
| Commercial assay or kit | LightCycler 480 SYBR Green I Master | Roche-Sigma-Aldrich | 04887352001 | N/A |
| Commercial assay or kit | Lipofectamine LTX PLUS reagent | Life Technologies | 15338–100 | N/A |
| Chemical compound, drug | Cardiotoxin | Latoxan | L8102 | 10 μM |
| Chemical compound, drug | bFGF | Peprotech | 450–33 | 20 ng/ml |
| Chemical compound, drug | Chicken embryo extract | MP-Biomedical | 2850145 | 0.5–1% |
| Chemical compound, drug | Collagenase A | Roche-Sigma-Aldrich | 10103586001 | 2 μg/ml |
| Chemical compound, drug | Collagenase type I | Sigma-Aldrich | C0130 | 0.2% |
| Chemical compound, drug | 4',6-diamidino-2-phenylindole dihydrochloride (DAPI) | Life Technologies | D1306 | N/A |
| Chemical compound, drug | Dispase II | Roche-Sigma-Aldrich | 10103586001 | 2.4 U/ml |
| Chemical compound, drug | DNaseI | Roche-Sigma-Aldrich | 1284932 | 10 ng/mL |
| Chemical compound, drug | Dulbecco's modified Eagle's medium (DMEM) | Life Technologies | 41966 | N/A |

*Continued*

| Reagent type (species) or resource | Designation | Source or reference | Identifiers | Additional information |
|---|---|---|---|---|
| Chemical compound, drug | DMEM with GlutaMAX | Life Technologies | 61965–026 | N/A |
| Chemical compound, drug | EdU | Thermo Fisher Scientific | C10340 | 2 µM |
| Chemical compound, drug | Fetal bovine serum (FBS) | Life Technologies | 10270 | 20% |
| Chemical compound, drug | Fluoromount-G | Southern Biotech | 0100–01 | N/A |
| Chemical compound, drug | Gelatin | Sigma-Aldrich | G1890 | 0.1% |
| Chemical compound, drug | Horse serum | Life Technologies | 26050088 | 5–10% |
| Chemical compound, drug | Penicillin/streptomycin | Life Technologies | 15140–122 | 1X |
| Chemical compound, drug | Tamoxifen | Sigma-Aldrich | T5648 | 5–10 µg/day |
| Software, algorithm | Metamorph Software | Molecular Devices | RRID: SCR_002368 | N/A |
| Software, algorithm | ImageJ | https://imagej.nih.gov/ij/ | RRID:SCR_003070 | N/A |

## Mice and animal care

*Pax3*$^{GFP/+}$ mouse strain was previously generated (*Relaix et al., 2005*). *Pax3*$^{Cre/+}$, *Pax7*$^{CreERT2/+}$ (*Pax7*$^{+/CE}$), *Tg:Pax7-nGFP*, and *Sox17* (*Sox17*$^{GFP/+}$ and *Sox17*$^{fl/+}$) mutant mice were kindly provided by Jonathan A. Epstein, Chen-Ming Fan, Shahragim Tajbakhsh and Sean J. Morrison, respectively (*Engleka et al., 2005*; *Kim et al., 2007*; *Lepper et al., 2009*; *Rocheteau et al., 2012*). All mice were maintained in a C56BL/6J background.

## Animal breeding

*Sox17*$^{fl/+}$ was inter-crossed to generate *Sox17*$^{fl/fl}$. *Sox17*$^{GFP/+}$ mice were bred with *Pax3*$^{Cre/+}$ in order to produce *Pax3*$^{Cre/+}$;*Sox17*$^{GFP/+}$ mutants, and the latter with *Sox17*$^{fl/fl}$ mice to obtain the ablation of *Sox17* in the muscle lineage (*Pax3*$^{Cre/+}$;*Sox17*$^{GFP/fl}$). For specific deletion of *Sox17* in satellite cells (*Pax7*$^{CreERT2/+}$;*Sox17*$^{fl/fl}$) *Sox17*$^{fl/fl}$ and *Pax7*$^{CreERT2/+}$ mice were crossbred. For recombination induction with the *Pax7CreERT2* allele, mice were fed in tamoxifen diet (TD.55125.I, Envigo) or intraperitoneally injected for four consecutive days in the adulthood (Roche-Sigma-Aldrich, St. Quentin Fallavier, France). Littermate *Sox17*$^{GFP/fl}$ or *Sox17*$^{fl/fl}$ were used as control animals (CTRL).

## Cell sorting and culture

For FACS, muscle samples were isolated from adult mice (forelimb, hindlimb, and trunk muscles). Following dissection, all muscles were minced and incubated in digestion buffer [HBSS (Life Technologies, Saint-Aubin, France), 0.2% BSA (Sigma-Aldrich, St. Quentin Fallavier, France), 2 µg/ml Collagenase A (Roche-Sigma-Aldrich, St. Quentin Fallavier, France), 2.4 U/ml Dispase II (Roche-Sigma-Aldrich, St. Quentin Fallavier, France), 10 ng/mL DNaseI (Roche-Sigma-Aldrich, St. Quentin Fallavier, France), 0.4 mM CaCl$_2$, and 5 mM MgCl$_2$], and purified by filtration using 100 µm and 40 µm cell strainers (BD Falcon, Le Pont de Claix, France). For labeling extracellular antigens, 10 ng/ml of the following antibodies were used: rat anti-mouse CD45-PE-Cy7 (BD, Le Pont de Claix, France), rat anti-mouse Ter119-PE-Cy7 (BD, Le Pont de Claix, France), rat anti-mouse CD34-BV421 (BD, Le Pont de Claix, France), rat anti-mouse integrin-α7-A700 (R and D Systems, Abingdon, UK), rat anti-mouse Sca1-FITC (BD, Le Pont de Claix, France), rat anti-mouse CD31-PE (BD, Le Pont de Claix, France). Muscle cells were stained using LIVE/DEAD® Fixable Blue Dead Cell Stain Kit (Life Technologies, Saint-Aubin, France) to exclude dead cells and purified via FACS Aria II based on TER119 (LY76)$^-$, CD45 (PTPRC, LY5)$^-$, CD34$^+$, SCA1$^-$ and gating on the cell fraction integrin-α7$^+$. Satellite cells

isolated from either *Pax3*$^{GFP/+}$ or *Tg:Pax7-nGFP* were obtained using the FITC channel to recover the GFP+ population.

Purified satellite cells were plated on 0.1% gelatin-coated dishes at low density for clonal analysis (500 cells/well in four-well plates). The remaining sorted cells were either frozen (quiescent) or plated for RNA extraction (proliferation or differentiation conditions). Cells were allowed to grow in proliferation medium: DMEM Glutamax containing 20% fetal bovine serum, 10% horse serum, 1% penicillin–streptomycin, 1% HEPES, 1% sodium pyruvate (Life Technologies, Saint-Aubin, France), 1/4000 bFGF (20 ng/ml Peprotech, Neuilly-sur-Seine, France) for one week at a density of 1000 cells/cm$^2$, and then switched into differentiation medium (5% HS) for four extra days.

## RNA preparation and quantitative PCR

Total RNA from FACS-sorted satellite cells was extracted from independent experiments according to the RNasy Micro Kit (QIAGEN, Courtaboeuf, France) RNA extraction protocol. For whole muscle total RNA, RNeasy Fibrous Tissue Midi Kit (QIAGEN, Courtaboeuf, France) was used. cDNA synthesis was performed using Transcriptor First Strand cDNA Synthesis Kit (Roche-Sigma-Aldrich, St. Quentin Fallavier, France). RNA quality was assessed by spectrophotometry (Nanodrop ND-1000).

qPCR reactions were carried out in triplicate using LightCycler 480 SYBR Green I Master (Roche-Sigma-Aldrich, St. Quentin Fallavier, France). Expression of each gene was normalized to that of Hypoxanthine Phosphoribosyltransferase 1 (*Hprt1*) for total muscle, or TATA Box Protein (*TBP*) for cultured cells. Results are given as mean ± standard error. The single (*), double (**), triple (***), and quadruple (****) asterisks represent *p*-values p<0.05, p<0.01 and p<0.001, respectively, for Student's unpaired t-test. The oligonucleotides used in this study are listed in *table 1*.

## Immunolabeling, microscopy and image treatment

Muscles were dissected and snap-frozen in liquid nitrogen-cooled isopentane. Eight µm cryosections were fixed in 4% paraformaldehyde (PFA) and immunofluorescence was carried out as previously described (*Mitchell et al., 2010*). Primary antibodies and used dilutions are summarized in *table 2*.

Secondary antibodies were Alexa 488 goat anti-mouse IgG (H + L), Alexa 546 goat anti-mouse IgG (H + L), Alexa 555 goat anti-mouse IgG (H + L), Alexa 594 goat anti-mouse IgG (H + L), Alexa 488 goat anti-rabbit IgG (H + L), Alexa 594 goat anti-rabbit IgG (H + L), Alexa 594 donkey anti-goat IgG (H + L), Alexa 488 goat anti-Chicken IgY (H + L) (Life Technologies, Saint-Aubin, France), and Cy5-goat anti-rabbit IgG (H + L) (Jackson ImmunoResearch, Suffolk, UK). Nuclei were counterstained with DAPI (Life Technologies, Saint-Aubin, France).

Analysis was carried out using either a Leica TCS SPE confocal microscope or a Zeiss AxioImager. Z1 ApoTome (for scanning of whole *Soleus* cryosections). Images were processed with either Adobe Photoshop CS5 software (Adobe Systems) or MetaMorph 7.5 Software (Molecular Devices). Counting was performed using ImageJ (version 1.47 v; National Institutes of Health, USA, https://imagej.nih.gov/ij/). Transduced satellite cells in myofiber cultures were directly counted under a Leica fluorescent microscope at 40x magnification. Mean ± standard error (s.e.m.) was given. The single (*), double (**), and triple (***) asterisks represent *p*-values p<0.05, p<0.01, and p<0.001 respectively by Student's unpaired t-test. All experiments have been performed on at least three independent experiments for each condition. For the characterization of *Sox17* mutant mice, 2–5 whole scanned cryosections in at least three different animals (controls and mutants) were analyzed.

## Single myofiber isolation, culture and transduction

Single myofiber procedure was performed as previously described (*Moyle and Zammit, 2014*). Briefly, both *Extensor digitorum longus* (EDL) muscles were dissected and digested in Collagenase type I (Sigma-Aldrich, St. Quentin Fallavier, France) solution for 1.5 hr. Flushing medium against the digested muscle, myofibers detached from whole muscle and were placed into another culture dish. Fibers were taken at different time points, freshly isolated (T0), and 24 (T24), 48 (T48), and 72 (T72) hours after culture in activation medium [DMEM High Glucose (Life Technologies, Saint-Aubin, France), 10% horse serum (Life Technologies, Saint-Aubin, France) and 0.5% chicken embryo extract (MP-Biomedical, Illkirch-Graffenstaden, France)] at 37°C in 5% CO$_2$. Retroviral expression vectors and transduction were carried out as previously reported (*Zammit et al., 2006*). To transduce myofiber-associated satellite cells, 1:10 dilution of the retroviral supernatant was used 24 hr after fiber

**Table 1.** List of primary antibodies used in this study for immunolabeling.

GFP, Green Fluorescent Protein; Ki67, Marker Of Proliferation Ki-67; MyoD1, Myogenic Differentiation 1; Pax7, Paired Box 7; Phospho-Histone H3 (Ser10), for detection of Histone H3 phosphorylated at serine 10; and Sox17, SRY-Box 17.

| Genes | Sequences |
|---|---|
| Pax7 | 5' – AGGCCTTCGAGAGGACCCAC – 3'<br>5' – CTGAACCAGACCTGGACGCG – 3' |
| Myf5 | 5' – TGAGGGAACAGGTGGAGAAC – 3'<br>5' – AGCTGGACACGGAGCTTTTA – 3' |
| Myod | 5' – GGCTACGACACCGCCTACTA – 3'<br>5' – GAGATGCGCTCCACTATGCT – 3' |
| Myog | 5' – AGTGAATGCAACTCCCACAG – 3'<br>5' – ACGATGGACGTAAGGGAGTG – 3' |
| Myh1 | 5' – CCAGGAGGCCCCACCCC – 3'<br>5' – CACAGTCCTCCCGGCCCC – 3' |
| Ki67 | 5' – CCTGTGAGGCTGAGACATGG – 3'<br>5' – TCTTGAGGCTCGCCTTGATG – 3' |
| Sox7 | 5' – CTTCAGGGGACAAGAGTTCG – 3'<br>5' – GGGTCTCTTCTGGGACAGTG – 3' |
| Sox17 | 5' – GCCAAAGACGAACGCAAGCGGT – 3'<br>5' – TCATGCGCTTCACCTGCTTG – 3' |
| Sox18 | 5' – AACAAAATCCGGATCTGCAC – 3'<br>5' – CGGTACTTGTAGTTGGGATGG – 3' |
| Ccnd1 | 5' – TTCCTCTCCTGCTACCGCAC – 3'<br>5' – GACCAGCCTCTTCCTCCACTTC – 3' |
| Jun | 5' – TCCCCTATCGACATGGAGTC – 3'<br>5' – TTTTGCGCTTTCAAGGTTTT – 3' |
| c-myc | 5' – GATTCCACGGCCTTCTCTCC – 3'<br>5' – GCCTCTTCTCCACAGACACC – 3' |
| Axin2 | 5' – AAGAGAAGCGACCCAGTCAA – 3'<br>5' – CTGCGATGCATCTCTCTCTG – 3' |
| Ppard | 5' – ATTCCTCCCCTTCCTCCCTG – 3'<br>5' – ACAATCCGCATGAAGCTCGA – 3' |
| Hprt1 | 5' – AGGGCATATCCAACAACAAACTT – 3'<br>5' – GTTAAGCAGTACAGCCCCAAA – 3' |
| TBP | 5' – ATCCCAAGCGATTTGCTG – 3'<br>5' – CCTGTGCACACCATTTTTCC – 3' |

DOI: https://doi.org/10.7554/eLife.26039.020

isolation. Satellite cells were transduced for 48 hr and then recovered for fixation and immunostaining. EdU (2 μM; C10340, Thermo Fisher Scientific, Montigny-le-Bretonneux, France) chase was performed for 72 hr (last 48 hr together with retroviral transduction). EdU-incorporating cells were detected according to the manufacturer's protocol.

## Retroviral cloning

*Sox7* and *Sox18* cDNAs were amplified by PCR from IMAGE clones 40131228 and 3967084 respectively; *Sox17* cDNA was cloned by PCR from mouse kidney cDNA (gift of Dr. J. Hadchouel). All were subcloned in *pCig* mammalian bi-cistronic expression vector and *pMSCV-IRES-eGFP* (MIG) retroviral packaging vector using XhoI and EcoRI added to cloning primers (*Megason and McMahon, 2002*; *Pear et al., 1998*).

## Muscle injury, electroporation, and histology

Control and mutant mice were injected with 40 μL of cardiotoxin (CTX; 10 μM, Latoxan, Portes-lès-Valence, France) in *Tibialis anterior* (TA) muscles following general anesthesia. Muscles were recovered 7, 10, and 28 days later, to compare control vs. mutant mice; for regeneration expression

**Table 2.** List of qPCR oligonucleotides used in this study

*Pax7*, Paired Box 7; *Myf5*, Myogenic Factor 5; *Myod1*, Myogenic Differentiation 1; *Myog*, Myogenin; *Myh1*, Myosin Heavy Chain 1; *Ki67*, Marker Of Proliferation Ki-67; *Sox7*, SRY-Box 7; *Sox17*, SRY-Box 17; *Sox18*, SRY-Box 18; *Ccnd1*, Cyclin D1; *Jun*, *Jun* Proto-Oncogene, AP-1 Transcription Factor Subunit; *c-myc*, MYC Proto-Oncogene, BHLH Transcription Factor; *Axin2*, Axin2; Ppard, Peroxisome Proliferator Activated Receptor Delta; *Hprt1*, Hypoxanthine Phosphoribosyltransferase 1; and *TBP*, TATA Box Protein.

| Antigen | Reference | Company | Ig type | Dilution |
|---|---|---|---|---|
| GFP | A11122 | Life Technologies | Rabbit IgG | 1:500 |
| GFP | ab13970 | Abcam | Chicken IgY | 1:500 |
| Ki67 | 556003 | BD Pharmingen | Mouse IgG1 | 1:100 |
| Ki67 | ab15580 | Abcam | Rabbit IgG | 1:100 |
| Laminin | L9393 | Sigma-Aldrich | Rabbit IgG | 1:100 |
| Laminin (AlexaFluor647) | NB300-144AF647 | Novus Biological | Rabbit IgG | 1:200 |
| M-Cadherin | MCAD-12G4 | nanoTools | Mouse IgG1 | 1:50 |
| MyoD1, 5.8A | M3512 | DAKO | Mouse IgG1 | 1:50 |
| MyoD, M-318 | sc-760 | Santa Cruz | Rabbit IgG | 1:20 |
| Myogenin | F5D | DSHB | Mouse IgG1 | 1:100 |
| Pax7 | PAX7-c | DSHB | Mouse IgG1 | 1:20 |
| Pax7 | sc-81648 | Santa Cruz | Mouse IgG1 | 1:20 |
| Phospho-Histone H3 (Ser10) | 06–570 | Merck Millipore | Rabbit IgG | 1:500 |
| Sox17 | AF1924 | R and D Systems | Goat IgG | 1:50 |

DOI: https://doi.org/10.7554/eLife.26039.021

profile, all days from day 0 up to day 7, and then days 10, 15, 21, and 28. Second injury was performed as above, 28 days after first injury. Muscle electroporation was performed using an Electro Square-Porator ECM 830 (BTX®, Genetronics Inc., Holliston, MA). According to (*Sousa-Victor et al., 2014*), 40 µg of DNA solutions were injected and TA muscles were electroporated using external plate electrodes two days after CTX injection. TAs were examined five, seven, or ten days later. Seven and 28 days after injury, TA muscles were processed for histology analysis by Hematoxylin and eosin , Oil red O, and Sirius red staining as previously described (*Sambasivan et al., 2011*).

## C2C12 culture and transfection for *β-galactosidase* reporter assays

C2C12 cells were grown in DMEM High Glucose (Life Technologies, Saint-Aubin, France) supplemented with 10% FBS (Bio West). A total of 1.2 µg DNA was transfected in $10^5$ cells using lipofectamine LTX PLUS reagent (Life Technologies, Saint-Aubin, France). Generated reporters were as follows: *SoxF-B-TKnLacZ*, five multimerized SOXF consensus binding motifs (annealed oligonucleotides 5'-CAACAATCATCATTGTTGGGGCCAACAATCTACATTGTTCAGA-3' and 5'-TCTGAACAATG TAGATTGTTGGCCCCAACAATGATGATTGTTG-3') (*Kanai et al., 1996*); β-catenin TOP *pTOP-TKnLacZ*, six tandem repeats of the TCF/LEF Transcriptional Response Element (*Molenaar et al., 1996*); *Ccnd1-nLacZ*, 1 kb of the 5'UTR region, encompassing binding motifs for TCF/LEF and SOXF proteins, was amplified from C57BL/6J genomic DNA (*Lange et al., 2009*); and *Axin2-nLacZ*, 5.6 kb of the proximal promoter fragment was excised from *Ax2-Luc* (gift of Dr. J. Briscoe) and subcloned (*Jho et al., 2002*). Fixed concentrations of all reporters (0.6 µg) were used. 48 hr after transfection, cells were lysed in 100 µl RIPA buffer supplemented with protease inhibitors (Complete Mini, Roche-Sigma-Aldrich, St. Quentin Fallavier, France). β-galactosidase assays were performed with 10 µl lysates based on 2-Nitrophenyl β-D-galactopyranoside (ONPG) substrate hydrolysis. When indicated, 1 mM LiCl treatment was performed 24 hr post-transfection and carried for 24 hr. Individual transfections were repeated at least three times; measurements are expressed as mean of the amount of ONPG hydrolyzed normalized to control. Error bars correspond to the standard error of the mean (s.e.m.). The single (*), double (**), triple (***), and quadruple (****) asterisks represent *p*-values p<0.05, p<0.01, p<0.001 and p<0.0001, respectively, for Mann-Whitney statistical test.

## Acknowledgements

We thank Sean J. Morrison for the *Sox17*$^{GFP/+}$ and *Sox17*$^{fl/+}$ mice, Jonathan A. Epstein for the *Pax3-$^{Cre/+}$* mice, Shahragim Tajbakhsh for the *Tg:Pax7-nGFP* mice, and Che-Ming Fan for the *Pax7*$^{CreERT2/+}$ mice; Edgar R. Gomes and Bruno Cadot for the C2C12 mouse cell line; James Briscoe for the *Ax2-Luc* construct and Juliette Hadchouel for providing the mouse kidney cDNA. The authors are grateful to Vanessa Ribes, Andrew TV Ho, Piera Smeriglio, and Maria Grazia Biferi for technical assistance and constructive comments; Peggy Lafuste and Zeynab Koumaiha for qPCR primers (*Ki67* and *Myh1*); Nora Butta, Raquel del Toro, Marta Flandez and Alysia vandenBerg for critical pre-submission review; Keren Bismuth, Ted Hung-Tse Chang and Bernadette Drayton for their input and assistance. We thank Catherine Blanc and Benedicte Hoareau (Flow Cytometry Core CyPS, Sorbonne Université, Pitié-Salpétrière Hospital), Adeline Henry and Aurélie Guguin (Plateforme de Cytométrie en flux, Institut Mondor de Recherche Biomédicale), and Serban Morosan and the animal care facility (Centre d'Expérimentation Fonctionnelle, Sorbonne Université). We finally want to thank the Histopathology and Microscopy Units at Centro Nacional de Investigaciones Cardiovasculares (CNIC, Spain). SA-M was recipient of a postdoctoral fellowship from the Basque Community (BF106.177). Funding from the German Research Society (DFG) through MyoGrad International Graduate School for Myology GK 1631 and KFO192 (Sp1152/8-1) and Labex REVIVE (ANR-10-LABX-73) supported DM. This work was further supported by funding to FR from INSERM Avenir Program, Association Française contre les Myopathies (AFM) via TRANSLAMUSCLE (PROJECT 19507), Association Institut de Myologie (AIM), Labex REVIVE (ANR-10-LABX-73), the European Union Sixth and Seventh Framework Program in the project MYORES and ENDOSTEM (Grant # 241440), Fondation pour la Recherche Médicale (FRM; Grant FDT20130928236 and DEQ20130326526), Agence Nationale pour la Recherche (ANR) grant Epimuscle (ANR 11 BSV2 017 02), Bone-muscle-repair (ANR-13-BSV1-0011-02), BMP-biomass (ANR-12-BSV1-0038- 04), Satnet (ANR-15-CE13-0011-01), BMP-MyoStem (ANR-16-CE14-0002-03), MyoStemVasc (ANR-17-CE14-0018-01), and RHU CARMMA (ANR-15-RHUS-0003). The lab of PSZ is supported by Muscular Dystrophy UK (RA3/3052), the Medical Research Council (MR/P023215/1), Association Française contre les Myopathies (AFM 17865 and AFM 16050), FSH Society (FSHS-82013-06 and FSHS-82016-03), and European Union Seventh Framework Program BIODESIGN (262948-2). The authors declare no competing financial interests.

## Additional information

### Funding

| Funder | Grant reference number | Author |
| --- | --- | --- |
| Basque Community | BF106.177 | Sonia Alonso-Martin |
| Deutsche Forschungsgemeinschaft | GK 1631 | Despoina Mademtzoglou |
| Deutsche Forschungsgemeinschaft | KFO192 (Sp1152/8-1) | Despoina Mademtzoglou |
| Horizon 2020 Framework Programme | MYORES and ENDOSTEM # 241440 | Peter S Zammit Frédéric Relaix |
| Muscular Dystrophy UK | RA3/3052 | Peter S Zammit |
| Medical Research Council | MR/PO23215/1 | Peter S Zammit |
| FSH Society | 262948-2 | Peter S Zammit |
| Horizon 2020 Framework Programme | BIODESIGN (262948-2) | Peter S Zammit |
| Association Française contre les Myopathies | AFM 17865 | Peter S Zammit |
| Association Française contre les Myopathies | AFM 16050 | Peter S Zammit |
| INSERM Avenir Program | | Frédéric Relaix |
| Association Française contre les Myopathies | TRANSLAMUSCLE 19507 | Frédéric Relaix |

| Association Institut de Myolo-gie | | Frédéric Relaix |
| --- | --- | --- |
| Labex REVIVE | ANR-10-LABX-73 | Frédéric Relaix |
| Fondation pour la Recherche Médicale | FDT20130928236 | Frédéric Relaix |
| Agence Nationale de la Re-cherche | ANR 11 BSV2 017 02 | Frédéric Relaix |
| Fondation pour la Recherche Médicale | DEQ20130326526 | Frédéric Relaix |
| Agence Nationale de la Re-cherche | ANR-13-BSV1-0011-02 | Frédéric Relaix |
| Agence Nationale de la Re-cherche | ANR-12-BSV1-0038-04 | Frédéric Relaix |
| Agence Nationale de la Re-cherche | ANR-15-CE13-0011-01 | Frédéric Relaix |
| Agence Nationale de la Re-cherche | ANR-15-RHUS-0003 | Frédéric Relaix |

The funders had no role in study design, data collection and interpretation, or the decision to submit the work for publication.

## Author contributions

Sonia Alonso-Martin, Conceptualization, Data curation, Formal analysis, Supervision, Funding acquisition, Validation, Investigation, Methodology, Writing—original draft, Writing—review and editing; Frédéric Auradé, Anne Rochat, Data curation, Formal analysis, Methodology, Writing—review and editing; Despoina Mademtzoglou, Data curation, Formal analysis, Funding acquisition, Methodology, Writing—review and editing; Peter S Zammit, Funding acquisition, Methodology, Writing—review and editing; Frédéric Relaix, Conceptualization, Resources, Formal analysis, Supervision, Funding acquisition, Validation, Investigation, Writing—original draft, Project administration, Writing—review and editing

## Author ORCIDs

Sonia Alonso-Martin iD http://orcid.org/0000-0002-3254-0365
Despoina Mademtzoglou iD http://orcid.org/0000-0002-4494-7234
Peter S Zammit iD http://orcid.org/0000-0001-9562-3072
Frédéric Relaix iD http://orcid.org/0000-0003-1270-1472

## Ethics

Animal experimentation: All animals were maintained inside a barrier facility and all experiment were performed in accordance with the European and French regulations for animal care and handling (Project No: 01427.03 approved by MESR and File No: 15-018 from the Ethical Committee of Anses/ENVA/UPEC).

## Decision letter and Author response

Decision letter https://doi.org/10.7554/eLife.26039.029
Author response https://doi.org/10.7554/eLife.26039.030

# Additional files

## Supplementary files

• Transparent reporting form
DOI: https://doi.org/10.7554/eLife.26039.022

## Data availability

Sequencing data have been deposited in GEO under accession code GSE63860 and previously published in: Gene Expression Profiling of Muscle Stem Cells Identifies Novel Regulators of Postnatal Myogenesis. Alonso-Martin S, Rochat A, Mademtzoglou D, Morais J, de Reyniès A, Auradé F, Chang TH, Zammit PS, Relaix F. Front Cell Dev Biol. 2016 Jun 21;4:58. doi: 10.3389/fcell.2016.00058. eCollection 2016. PMID: 27446912.

The following previously published dataset was used:

| Author(s) | Year | Dataset title | Dataset URL | Database, license, and accessibility information |
|---|---|---|---|---|
| Alonso-Martin S, Rochat A, de Reyniès A, Relaix F | 2016 | Chronological expression data from mouse skeletal muscle stem cells | https://www.ncbi.nlm.nih.gov/geo/query/acc.cgi?acc=GSE63860 | Publicly available at the NCBI Gene Expression Omnibus (accession no: GSE63860). |

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
