## [Decision Letter]

Thank you for submitting your article "SOXF factors regulate satellite cell self-renewal and function through inhibition of β-catenin activity" for consideration by *eLife*. Your article has been evaluated by Fiona Watt (Senior Editor) and three reviewers, one of whom, Amy J Wagers (Reviewer #1), is a member of our Board of Reviewing Editors. The following individual involved in review of your submission has also agreed to reveal their identity: Pier Lorenzo Puri (Reviewer #3).

The reviewers have discussed the reviews with one another and the Reviewing Editor has drafted the following summary and review.

Summary:

This manuscript identifies SOXF (SOX7, SOX17, and SOX18) members as highly expressed transcription factors in adult *Pax3^GFP^*^+^ cells and evaluates the role of SOX17 in regulating satellite cell fate through overexpression and loss of function studies using *Pax3^Cre^-Sox17*null mice and electroporation of *Sox17* dominant negative constructs. The authors also demonstrate potential interactions between SOXF family members and an effector of canonical Wnt signaling, b-catenin.

This paper investigates the role of *Sox7, Sox17* and *Sox18* (cumulatively, *SoxF*) in adult muscle satellite cell biology. The authors report developmentally regulated expression of *SoxF* in muscle and, and further assess the impact of *SoxF* gain- and loss-of-function on muscle satellite cells *in vivo* and *in vitro* using transfection studies and gene-modified mice. They conclude from these studies that *SoxF* regulate satellite cell "quiescence, self-renewal and differentiation". They further demonstrate significant impairment of muscle regenerative function in animals lacking *SoxF (Sox17*) function in muscle. Finally, they perform a series of reporter assays that suggest an interaction of *SoxF* with Wnt signaling (β-catenin).

Overall, the work is intriguing for its potential identification of a new class of muscle satellite cell regulators, and particularly the suggestion that *SoxF*'s regulate satellite cell quiescence, a state whose regulation is poorly understood. Still, there are a number of places where the reviewers raised concerns that authors' conclusions may overreach the available data. These are outlined below, and would need to be addressed via new experimentation and revision of the manuscript text.

Essential revisions:

1) The data in Figure 1 were obtained from sorted *Pax3^GFP^* cells, where GFP is a reporter for *Pax3* expression, not a lineage tracer (Relaix et al., 2006). For adult satellite cell collection this is a problem since, based on the authors' previously published work, *Pax3* expression is rare, except in the diaphragm, in adult satellite cells. Furthermore, *Pax3^GFP^* expression is observed in major adult blood vessels in the limbs (Goupille et al., 2011). Based on the Materials and methods (subsection “Cell sorting and culture”, first paragraph), CD31 is not used for negative selection. Since hindlimb, forelimb and trunk muscles were used one cannot be sure what fraction of the cells in Figure 1 are CD31+/*Pax3^GFP^*^+^ blood vessel derived cells. Due to the questions with regards to *Pax3^GFP^* expression and lack of CD31 use, sort profiles with all gates and populations need to be shown to demonstrate how *Pax3^GFP^*^+^ cells were prospectively isolated. Also, they need to report what proportion of the isolated *Pax3^GFP^*^+^ cells used for their studies express *Pax7* and what proportion express *Sca1* (by immunostaining of isolated cells and FACS), clarify if they used CD31 counterselection in their sorts, and assess the anatomical localization of the cells (sublaminar vs. interstitial) by staining in tissue sections.

2) Throughout the manuscript, the authors often inappropriately conflate self-renewal and quiescence with *Pax7* positivity. For example, in reporting the results of retroviral transductions in Figure 2, they claim that overexpression of *SoxF* promotes "self-renewal", but their data show changes in the frequency of *Pax7*^+^ and *MyoD*^+^ cells and does not assay self-renewal itself (which would require tracking of either individual cell divisions or minimally of input cell number versus output cell number). Similarly, in Figure 3, the authors have not directly evaluated quiescence, and in Figures 5 and 6, they have not evaluated self-renewal (just *Pax7* expression). Finally, the authors' conclusion that "Our findings point to modulation of cell cycle by SOXF activity: satellite cells fail to acquire quiescence when SOXF function is impaired *in vivo* and *ex vivo*" is problematic, as in several studies the authors have not directly assessed proliferation or quiescence states, and have only inferred these from *pax7/myod* expression. For renewal, the authors need to evaluate absolute numbers of *Pax7*^+^, *MyoD*^+^, and *Myog*^+^ cells per fiber. EdU pulse experiments would be helpful as well. An alternative explanation for the *Sox7FL* effects could be decreased activation as opposed to supporting renewal. Based on the *Sox7DN* data, are the authors suggesting that loss of *Pax7* is coupled to premature differentiation or apoptosis? At 72 hours *Myog*^+^ cells are observed yet their proportion does not increase. Depending on the absolute number count per fiber the authors will probably also need to assess apoptosis.

3) A major limitation of the authors' conclusions, which center entirely around the notion that *SoxF*'s exert their influence on muscle repair capacity by regulating transcriptional events in satellite cells themselves is that neither of the models they use to evaluate the *in vivo* regenerative phenotypes caused by loss of *SoxF*'s are satellite cell specific. For example, the data in Figure 3 are generated from *Pax3^Cre^/Sox17^fl^*(*Sox17* KO) mice. Although *Sox17* is not abundantly expressed in embryonic *Pax3^GFP^*+ cells (Figure 1A and 1B), the Cre is and therefore all cells derived from the *Pax3* lineage will be disrupted for *Sox17*. Such *Sox17* disrupted cells in addition to satellite cells and derived myogenic progenitors include myonuclei and blood vessel cells(Goupille et al., 2011; Relaix et al., 2006). Therefore, the muscle fiber disruptions such as fiber type transitions and atrophy could reflect *Sox17* roles in muscle fibers and blood vessels. Thus, the authors cannot exclude potential influences of *SoxF* loss in other cell types, and should address this issue using an inducible satellite cell specific model (e.g., *Pax7^CreER^*).

4) The manner of myonuclei count (Figure 3F) is not adequate, the authors should obtain dissociated single fibers and count myonuclei along their length (Brack et al., 2005). The assumption is that myonuclear loss reflects loss of satellite cells and derived fusion competent myogenic progenitors. Also, no assessment of myogenic progenitor number or activity (BrdU/EdU) is done at any stage up to adult skeletal muscle ages. The reduction in Pax7 numbers in Figure 3H and loss of quiescence in Figure 3I could reflect disruptions in the muscle fiber niche (loss of *Sox17* in myonuclei or blood vessel cells).

5) It is unclear whether the fiber count and sizing of soleus sections in this figure are valid given the number of fibers cut longitudinally in the representative images. It seems that the results could be skewed if interpreting sections with these artifacts. Also, TA and EDL muscle data should also be included or mentioned since they use these muscles for regeneration experiments and culture.

6) Figure 4 should include analysis by immunofluorescence with appropriate fate markers and/or FACs of satellite cell and derived myogenic progenitor numbers at stages of regeneration where fate decisions are readily apparent (~3-7 days after injury). To assess renewal and proliferation, BrdU or EdU pulse experiments with appropriate fate markers should be conducted. Considering *Sox17* would be lost in *Pax3* derived cells (myonculei and blood vessels), it is difficult to comprehend the conclusion that the phenotypes strictly reflect satellite cell autonomous fate decisions.

7) For the *in vivo* electroporation studies (Figure 5D-F), it is important to evaluate GFP expression at early and late time points to assess the transfection efficiency and document the cell types in which the dominant negative *SoxF* is expressed. Also, pockets of ORO+ and Sirius red+ reactivity seem regional in these tissues. How were regions chosen for the quantification shown in this figure?

8) A physical interaction needs to be demonstrated between b-catenin and SOXF family members to support the authors' conclusions. Also, whether these interactions are lost upon removal of the b-catenin interaction site in the SOXF family needs to be tested. Considering the timing of canonical Wnt activity and SOXF family member expression during regeneration, it is not immediately clear as to the relevance of SoxF factors with b-catenin in the context of this manuscript (Brack et al., 2008; Murphy et al., 2014; Rudolf et al., 2016). Although controversial, b-catenin activity is highest during stages of active fate decisions in satellite cells and myogenic progenitors during regeneration (days 3-7 after injury). Yet in Figure 4, *SoxF* expression peaks at later stages of regeneration. Also, these transcripts are measured in whole TA muscle, whereas they should be measured in sorted satellite cells and myogenic progenitors. Alternatively, SoxF family members along with myogenic fate markers *Pax7, MyoD*, and *Myog* could be tested with immunofluorescence at days 3-7 during injury. Since based on the literature and the data in this study, b-catenin activity is associated with myogenic progression, and *SoxF* family members are proposed to interfere with b-catenin activity; the authors should test localization of *SoxF* members (*Sox17*) with myogenic fate regulators at days 3-7 of muscle regeneration. Another possibility is b-catenin activity should be higher in *Sox17* null mice, which could explain some of the phenotypes observed in this manuscript (Murphy et al., 2014).

9) Figure 6C – β-catenin target gene analyses lack statistical assessment. Also, it appears that the b-cat target genes show variable differences in the *Sox17* deficient muscles. The analysis is also complicated by the different cellular and fiber type composition of the muscles in the *Sox17* deficient animals. These issues should be accounted for in the authors' presentation and interpretation of these results.

10) Figure 7E*Axin2* expression is assayed in sorted *Pax3^GFP^*^+^ cells without CD31 negative selection this is problematic based on the authors publications as described above (Goupille et al., 2011; Relaix et al., 2006). Due to the questions with regards to *Pax3^GFP^* expression and lack of CD31 use, sort profiles with all gates and populations need to be shown to demonstrate how *Pax3^GFP^*^+^ cells were prospectively isolated.

11) LiCl is a GSK3b inhibitor, and so, as GSK3b has additional activities that are not related to its role in Wnt signaling, LiCl should not be presented as a specific Wnt activator.

12) The authors need to add an important control to the studies comparing *SoxF* overexpression and ability to rescue – they must assess the level of overexpression of the various *Sox7, Sox17* and *Sox18* constructs and ensure that they are similarly overexpressed. Similarly, they provide no evidence that endogenous SOXF protein levels parallel the transcript levels of *SoxF* genes. These points need to be addressed by the authors.

13) The authors described an increase of slow fibers in soleus muscle of *Sox17^fl^/Pax3^Cre^* mice. First, a more complete analysis of fast and slow myosins should be performed in various muscles should be performed. Second, the authors should at least discuss the potential connection between SOXF expression and muscle metabolism.

[Editors' note: further revisions were requested prior to acceptance, as described below.]

Thank you for resubmitting your work entitled "SOXF factors regulate satellite cell self-renewal and function through inhibition of β-catenin activity" for further consideration at *eLife*. Your revised article has been favorably evaluated by Fiona Watt (Senior Editor), a Reviewing Editor, and two reviewers.

The manuscript has been improved but one of the reviewers has raised a few remaining issues that need to be addressed before acceptance, as outlined below:

In response to the first review the authors submitted a figure demonstrating lack of *Pax3^Cre-GFP^* in CD31^+^ cells. Inclusion of this as a supplement would be helpful.

It is impressive that the authors observe a similar magnitude of *Pax7*^+^ SC loss regardless of whether *Pax3^Cre^* or *Pax7^CreER^* is used. However, there are discrepancies in the regeneration experiments. The *Pax3^Cre^-Sox17* KO mice demonstrate obvious impairments in regeneration (Figure 5) after 28 days of recovery. Figure 6 demonstrates *Pax7*^+^ cell loss in *Pax7^CreER^-Sox17* KO mice after only 7 days of recovery. There are no data from 28 day regenerated *Pax7^CreER^-Sox17* KO muscle. Therefore it is not known whether regeneration or satellite cell renewal are impaired after 28 days in the *Pax7^CreER^-Sox17* KO. These data should be provided and compared/discussed with the *Pax3^Cre^* data/published studies to determine if differences in regenerative phenotype occur depending on timing of recombination. Also, some regenerative measures should be quantified for example size of regenerated muscle fibers,% Oil red O area, and% Sirius red area.

---

## [Author Response]

1) The data in Figure 1 were obtained from sorted Pax3^GFP^ cells, where GFP is a reporter for Pax3 expression, not a lineage tracer (Relaix et al., 2006). For adult satellite cell collection this is a problem since, based on the authors' previously published work, Pax3 expression is rare, except in the diaphragm, in adult satellite cells.

*Pax3* expression in adult has been previously reported in Relaix et al. (2006) and Calhabeu et al. (2012). *Pax3* expression is, as noted by the reviewer, restricted in many muscles such as hindlimb muscles but expressed in a large number of body muscles, with a variable proportion of satellite cells in a muscle-specific pattern. Most of the trunk muscles express PAX3 for instance. Accordingly, data presented in Figure 1A and 1B were obtained from sorted trunk (not diaphragm) *Pax3^GFP^*^+^ cells. FACS sorting and dissections were optimized at all stages to avoid contamination from other *Pax3*-expressing cells, such as melanocytes in the fetal stages. *Pax3*-GFP+ FACS sorting strategy has previously been optimized during early myogenic development (Lagha et al., 2010) and adult satellite cells (Pallafacchina et al., 2010). This screen merely identifies interesting target genes, which were then examined in more detail to confirm and extend expression dynamic studies in satellite cells in general. Data shown in Figure 1D (including culture of satellite cells) were obtained from sorted TER119 (LY76)^-^, CD45 (PTPRC, LY5)^-^, SCA1^-^, CD34^+^, and integrin-α7^+^ from forelimb, hindlimb, and trunk (not diaphragm) adult muscles.

Calhabeu, F., Hayahi, S., Morgan, J.E., Relaix, F. and Zammit, P. 2012 Alveolar Rhabdomyosarcoma-associated proteins PAX3/FOXO1A and PAX7/FOXO1A 2 suppress the transcriptional activity of MyoD-target genes in muscle stem cells. Oncogene. 32(5):651-62.

Lagha M, Sato T, Regnault B, Cumano A, Zuniga A, Licht J, Relaix F, Buckingham M. 2010. Transcriptome analyses based on genetic screens for Pax3 myogenic targets in the mouse embryo. BMC Genomics 11: 696.

Pallafacchina G, François S, Regnault B, Czarny B, Dive V, Cumano A, Montarras D, Buckingham M. 2010. An adult tissue-specific stem cell in its niche: a gene profiling analysis of *in vivo* quiescent and activated muscle satellite cells. Stem Cell Res 4: 77.

Relaix, F., Montarras, D., Zaffran, S., Gayraud-Morel, B., Rocancourt, D., Tajbakhsh, S., Mansouri, A., Cumano, A., and Buckingham, M. (2006). Pax3 and Pax7 have distinct and overlapping functions in adult muscle progenitor cells. J. Cell Biol., 172, 91-102.Calhabeu 2012 Oncongene

Furthermore, Pax3^GFP^ expression is observed in major adult blood vessels in the limbs (Goupille et al., 2011). Based on the Materials and methods (subsection “Cell sorting and culture”, first paragraph), CD31 is not used for negative selection. Since hindlimb, forelimb and trunk muscles were used one cannot be sure what fraction of the cells in Figure 1 are CD31+/Pax3^GFP+^ blood vessel derived cells.

In adult muscle, *Pax3* expression is restricted to muscle satellite cells; no expression is seen in either blood vessels or neural crest-derived lineages. Goupille et al. (2011) demonstrate the presence of PAX3 in smooth muscle cells (mural cells) from peripheral arteries, which we avoid during dissection. Furthermore, that paper shows that the blood-vessel-derived cells only rarely and non-cell autonomously contribute to muscle fiber formation; their myogenic potential requires co-culture with skeletal muscle cells and cell fusion. More importantly, in Figure 3C of Goupille et al. (2011), the authors demonstrate that CD31 expression is restricted to the GFP- population and it is not in the GFP+ population. When performing *Pax3^GFP^*/CD31 FACS analysis, we found hardly any overlap between the *Pax3^GFP^* and CD31 immunolabeling (i.e. 1% or less). These data are presented in Figure 1—figure supplement 1 of the published manuscript.

Goupille O, Pallafacchina G, Relaix F, Conway SJ, Cumano A, Robert B, Montarras D, Buckingham M. 2011. Characterization of Pax3-expressing cells from adult blood vessels. J Cell Sci 124: 3980.

Due to the questions with regards to Pax3^GFP^ expression and lack of CD31 use, sort profiles with all gates and populations need to be shown to demonstrate how Pax3^GFP+^ cells were prospectively isolated. Also, they need to report what proportion of the isolated Pax3^GFP+^ cells used for their studies express Pax7 and what proportion express Sca1 (by immunostaining of isolated cells and FACS), clarify if they used CD31 counterselection in their sorts, and assess the anatomical localization of the cells (sublaminar vs. interstitial) by staining in tissue sections.

This *Pax3* expression point was addressed in our latest publication (Alonso-Martin et al., 2016), where we performed a genome-wide chronological expression profile from similar PAX3+ sorted cells. All FACS profiles are indeed illustrated in Alonso-Martin et al., 2016 – Figure S2A. Moreover, to exclude a possible contamination of satellite cells with endothelial cells, we performed PAX3-lineage tracing using *Pax3^Cre/+^;R26^mTmG^* mice (Alonso-Martin et al., 2016 – Figure S2B). While adult myogenic cells were mGFP+ (*Pax3-Cre* recombined), all endothelial cells remained mTOMATO+ (not recombined) (Alonso-Martin et al., 2016 – Figure S2B and Movie S1). More importantly, all CD31(PECAM-1)+ endothelial cells were included within the mTOMATO+ population (Alonso-Martin et al., 2016 – Figure S2B and Movie S2). These results demonstrate that the PAX3 lineage does not contribute to skeletal muscle endothelial population, and that skeletal muscle expression of PAX3 is specific to muscle stem cells. Finally, since PAX3 is expressed in a subset of the PAX7-expressing satellite cells, we compared our gene expression data with previously published datasets of adult muscle stem cells where markers different from PAX3 were used to isolate satellite cells (Alonso-Martin et al., 2016 – Figure S3A). Liu et al., 2013: VCAM+CD31-CD45-SCA1- or the YFP fraction from *Pax7^CreERT2/+^;ROSA26^eYFP/+^*; Sinha et al., 2014: CD45-TER119-SCA1-CD29+CXCR4. PAX3-expressing satellite cells were not significantly divergent from previously reported datasets. Moreover, we compared available data from adult (3-8 month-old) and old satellite cells (18-24 month-old) with our data, identifying a similar variation in all datasets. Finally, this was also addressed in reviewer Figure 1 above (and see previous response).

Liu L, Cheung TH, Charville GW, Hurgo BM, Leavitt T, Shih J, Brunet A, Rando TA. 2013. Chromatin modifications as determinants of muscle stem cell quiescence and chronological aging. Cell Rep 4: 189.

Sinha M, Jang YC, Oh J, Khong D, Wu EY, Manohar R, Miller C, Regalado SG, Loffredo FS, Pancoast JR, Hirshman MF, Lebowitz J, Shadrach JL, Cerletti M, Kim MJ, Serwold T, Goodyear LJ, Rosner B, Lee RT, Wagers AJ. 2014. Restoring systemic GDF11 levels reverses age-related dysfunction in mouse skeletal muscle. Science 344: 649.

2) Throughout the manuscript, the authors often inappropriately conflate self-renewal and quiescence with Pax7 positivity. For example, in reporting the results of retroviral transductions in Figure 2, they claim that overexpression of SoxF promotes "self-renewal", but their data show changes in the frequency of Pax7^+^ and MyoD^+^ cells and does not assay self-renewal itself (which would require tracking of either individual cell divisions or minimally of input cell number versus output cell number). Similarly, in Figure 3, the authors have not directly evaluated quiescence, and in Figures 5 and 6, they have not evaluated self-renewal (just Pax7 expression).

In single myofiber cultures (Figure 2, 5, 6 of initial manuscript), PAX7 positivity was used as a proxy for self-renewal, as it was previously shown that self-renewing cells in this system are PAX7+ (Zammit et al., 2004). In addition, we show that SOXF not only increase PAX7, but concomitantly reduces the proportion of cells with KI67 (so in the cell cycle), and importantly, also the proportion expressing myogenin (Figure 2 and Figure 2—figure supplement 1). Thus a higher proportion of PAX7 cells are exiting the cell cycle but not differentiating, and so self-renewal is the most obvious interpretation of this data. This interpretation is strengthened by the *in vivo* quantification showing less satellite cells and more with MYOD expression in both *Pax3^Cre/+^;Sox17^GFP/fl^*and the new *Pax7^CreERT2/+^;Sox17^fl/fl^* model(Figures 3 and 4).

Regarding cycling status in Figure 3 (now Figure 4), adult satellite cells are known to have exited the cell cycle (White et al., 2010), while for P14 additional staining of PAX7/PH3 was performed, showing a tendency for increased proliferation in mutant satellite cells (Author response image 1).

**Author response image 1. respfig1:** Evaluation of cycling and self-renewal status. Quantification of the cycling (PH3+) satellite cells (PAX7+) at P14. Data expressed as mean ± s.e.m.

Finally, the authors' conclusion that "Our findings point to modulation of cell cycle by SOXF activity: satellite cells fail to acquire quiescence when SOXF function is impaired *in vivo* and *ex vivo*" is problematic, as in several studies the authors have not directly assessed proliferation or quiescence states, and have only inferred these from pax7/myod expression.

When quiescent satellite cells are stimulated for growth, MYOD expression is one of the hallmarks of their activation. MYOD protein presence has been reported in myoblasts of single myofibers as early as 2 hours following culture (Zhang et al., 2010), before even re-entry to cell cycle and first division, which occurs at 24-48 hours based on live imaging experiments (Siegel et al., 2011). Thus, evaluating PAX7/MYOD provides an indirect way to assess proliferation and quiescence.

Siegel AL, Kuhlmann PK, Cornelison DD. Muscle satellite cell proliferation and association: new insights from myofiber time-lapse imaging. 2011. Skelet Muscle 1: 7.

Zhang K, Sha J, Harter ML. 2010. Activation of Cdc6 by MyoD is associated with the expansion of quiescent myogenic satellite cells. J Cell Biol 188: 39.

For renewal, the authors need to evaluate absolute numbers of Pax7^+^, MyoD^+^, and Myog^+^ cells per fiber. EdU pulse experiments would be helpful as well. An alternative explanation for the Sox7FL effects could be decreased activation as opposed to supporting renewal. Based on the Sox7DN data, are the authors suggesting that loss of Pax7 is coupled to premature differentiation or apoptosis? At 72 hours Myog^+^ cells are observed yet their proportion does not increase. Depending on the absolute number count per fiber the authors will probably also need to assess apoptosis.

As suggested by the reviewers, EdU pulse experiments have been performed and the results are reported in Figure 2—figure supplement 1B. For these experiments, we produced new viruses in which the fluorescent tracker is restricted to the endoplasmic reticulum and was visualized by a secondary antibody in the blue channel. Myofibers were cultured in the presence of EdU (from T0 to T72) and transduced (CFP+) satellite cells were screened for EdU presence at T72. >98% of CFP+ cells were EdU+, demonstrating that the transduced cells that we quantify at T72 represent progeny of activated satellite cells and not quiescent satellite cells that failed to activate (Zammit et al., 2004).

Apoptosis is difficult to estimate in the floating myofiber system, because apoptotic cells detach from the fibers. Instead, TUNEL assay was performed in cryosections of d7 regenerating muscles of mutant and control mice, not revealing significant differences.

3) A major limitation of the authors' conclusions, which center entirely around the notion that SoxF's exert their influence on muscle repair capacity by regulating transcriptional events in satellite cells themselves is that neither of the models they use to evaluate the *in vivo* regenerative phenotypes caused by loss of SoxF's are satellite cell specific. For example, the data in Figure 3 are generated from Pax3^Cre^/Sox17^fl^ (Sox17 KO) mice. Although Sox17 is not abundantly expressed in embryonic Pax3^GFP^+ cells (Figure 1A and 1B), the Cre is and therefore all cells derived from the Pax3 lineage will be disrupted for Sox17. Such Sox17 disrupted cells in addition to satellite cells and derived myogenic progenitors include myonuclei and blood vessel cells(Goupille et al., 2011; Relaix et al., 2006). Therefore, the muscle fiber disruptions such as fiber type transitions and atrophy could reflect Sox17 roles in muscle fibers and blood vessels. Thus, the authors cannot exclude potential influences of SoxF loss in other cell types, and should address this issue using an inducible satellite cell specific model (e.g., Pax7^CreER^).

As suggested by the reviewers, we generated satellite-cell specific inducible animals by combining *Pax7-CreERT2* and *Sox17flox* alleles. Data presented in Figure 4E-G and Figure 3—figure supplement 2 show that in the resting *Soleus* muscle, ablation of *Sox17* in satellite cells leads to loss of PAX7+ cells while having minimal impact on muscle structure 21 days after TMX injection. Moreover, we performed regeneration 15 days after the last TMX injection. TA muscles were injured with CTX and analyzed 7 days post-injury (Figure 6). *Pax7^CreERT2^;Sox17^fl/fl^* mice display similar phenotype to that observed in *Pax3^Cre/+^;Sox17^fl/fl^*, namely increased cell infiltration and fibrosis, and more importantly, loss of PAX7+ cells. Together, our data confirm the role of SOXF specifically in satellite cells.

4) The manner of myonuclei count (Figure 3F) is not adequate, the authors should obtain dissociated single fibers and count myonuclei along their length (Brack et al., 2005).

Both methodologies provide valid data on the number of myonuclei, but only one method needs to be performed to determine the number in our opinion. In accordance with previous reports (reference list below), we chose to quantify myonuclei in cross-sections so that a) results would not be affected by possible differences in individual fibers' length, b) the thousands of fibers of the entire muscle would not be under-represented by counting only 30-40 isolated myofibers. We thank the reviewers for their comment and indeed acknowledge that the quantification of cross-sections is more representative of myonuclei density. We therefore clarified the text accordingly.

Reference List

Adams GR, Caiozzo VJ, Haddad F, Baldwin KM. 2002. Cellular and molecular responses to increased skeletal muscle loading after irradiation. Am J Physiol Cell Physiol 283: C1182.

Bruusgaard JC, Johansen IB, Egner IM, Rana ZA, Gundersen K. 2010. Myonuclei acquired by overload exercise precede hypertrophy and are not lost on detraining. Proc Natl Acad Sci 107: 15111.

Egner IM, Bruusgaard JC, Eftestøl E, Gundersen K. 2013. A cellular memory mechanism aids overload hypertrophy in muscle long after an episodic exposure to anabolic steroids. J Physiol 591: 6221.

Kadi F, Schjerling P, Andersen LL, Charifi N, Madsen JL, Christensen LR, Andersen JL. 2004. The effects of heavy resistance training and detraining on satellite cells in human skeletal muscles. J Physiol 558: 1005.

Karlsen A, Couppé C, Andersen JL, Mikkelsen UR, Nielsen RH, Magnusson SP, Kjaer M, Mackey AL. 2015. Matters of fiber size and myonuclear domain: Does size matter more than age? Muscle Nerve 52: 1040.

Kirby TJ, McCarthy JJ, Peterson CA, Fry CS. 2016. Synergist ablation as a rodent model to study satellite cell dynamics in adult skeletal muscle. Methods Mol Biol 1460:43.

Kirby TJ, Patel RM, McClintock TS, Dupont-Versteegden EE, Peterson CA, McCarthy JJ. 2016. Myonuclear transcription is responsive to mechanical load and DNA content but uncoupled from cell size during hypertrophy. Mol Biol Cell 27: 788.

Liu F, Fry CS, Mula J, Jackson JR, Lee JD, Peterson CA, Yang L. 1985. Automated fiber-type-specific cross-sectional area assessment and myonuclei counting in skeletal muscle. J Appl Physiol 115: 1714.

McLoon LK, Rowe J, Wirtschafter J, McCormick KM. 2004. Continuous myofiber remodeling in uninjured extraocular myofibers: myonuclear turnover and evidence for apoptosis. Muscle Nerve 29: 707.

Merrick D, Stadler LK, Larner D, Smith J. 2009. Muscular dystrophy begins early in embryonic development deriving from stem cell loss and disrupted skeletal muscle formation. Dis Model Mech 2: 374.

Schwartz LM, Brown C, McLaughlin K, Smith W, Bigelow C. 2016. The myonuclear domain is not maintained in skeletal muscle during either atrophy or programmed cell death. Am J Physiol Cell Physiol 311: C607.

The assumption is that myonuclear loss reflects loss of satellite cells and derived fusion competent myogenic progenitors. Also, no assessment of myogenic progenitor number or activity (BrdU/EdU) is done at any stage up to adult skeletal muscle ages. The reduction in Pax7 numbers in Figure 3H and loss of quiescence in Figure 3I could reflect disruptions in the muscle fiber niche (loss of Sox17 in myonuclei or blood vessel cells).

PAX7+ satellite cells at post-natal day 14 (P14) were assessed for PH3 expression. The mutants showed a tendency for increased proliferation, although in a non-statistically significant manner. Results are reported in Author response image 1. Moreover, *Sox17* is not expressed in the myonuclei of the myofibers in single fiber cultures (Figure 1C) and Pax3-Cre is not expressed in the vascular cells (Figure 1—figure supplement 1 of the published manuscript). We are therefore confident that the reduction in PAX7 number is due to loss of SOX17 in the satellite cells of Pax3-Cre:Sox17flox mice, especially considering the similar observations using the new *Pax7^CreERT2/+^;Sox17^fl/fl^*model.

5) It is unclear whether the fiber count and sizing of soleus sections in this figure are valid given the number of fibers cut longitudinally in the representative images. It seems that the results could be skewed if interpreting sections with these artifacts.

We wished to illustrate a whole soleus muscle in cross section to provide an overview to show that the reduction in muscle fiber size was not restricted to certain areas/regions. This was one of N≥4 mice analyzed in triplicate where total number of fibers per whole muscle cross section was counted and approximately the CSA of 1500-2000 myofibers measured. No longitudinally cut fibers were found in the images used for fiber sizing.

Also, TA and EDL muscle data should also be included or mentioned since they use these muscles for regeneration experiments and culture.

We thank the reviewers for this suggestion. We performed as requested an analysis on other muscles. Analyzing further muscles showed that the situation is more complex than anticipated, with fast and slow muscles being affected differentially (Author response image 2). We will further investigate this phenotype in future studies and so we have currently removed the relevant data from Figure 3.

**Author response image 2. respfig2:** Effect of *Sox17* deletion on myofiber type distribution. Quantification of the slow-type MyHCI+ myofibers expressed as percentage of all fibers in whole adult *Soleus* (**A**), TA (**B**), and EDL (**C**) cross-sections from control and *Sox17* mutant mice. CTRL, *Sox17^GFP/fl^*; KO, *Pax3^Cre/+^;Sox17^GFP/fl^*. n≥4 mice (each in triplicate) for all experiments. Data expressed as mean ± s.e.m, statistically analyzed with Student’s unpaired t-test: ***, *p*<0.001.

6) Figure 4 should include analysis by immunofluorescence with appropriate fate markers and/or FACs of satellite cell and derived myogenic progenitor numbers at stages of regeneration where fate decisions are readily apparent (~3-7 days after injury). To assess renewal and proliferation, BrdU or EdU pulse experiments with appropriate fate markers should be conducted. Considering Sox17 would be lost in Pax3 derived cells (myonculei and blood vessels), it is difficult to comprehend the conclusion that the phenotypes strictly reflect satellite cell autonomous fate decisions.

We FACS-isolated satellite cells before (d0) and during regeneration (d2-d5-d7) and performed qRT-PCR analysis on SoxF genes and fate markers using the reporter *Tg:Pax7-nGFP* mice (Rocheteau et al., 2012). These data have been added in revised Figure 5 and revised Figure 5—figure supplement 1. New data from the new *Pax7^CreERT2/+^;Sox17^fl/fl^*model also confirms that satellite cells are clearly perturbed when *Sox17* is deleted in just that cell population with reduced Pax7 cell numbers, but a higher proportion still in cell cycle (Figure 6).

7) For the *in vivo* electroporation studies (Figure 5D-F), it is important to evaluate GFP expression at early and late time points to assess the transfection efficiency and document the cell types in which the dominant negative SoxF is expressed. Also, pockets of ORO+ and Sirius red+ reactivity seem regional in these tissues. How were regions chosen for the quantification shown in this figure?

New electroporation experiments were performed for early (d5) and late (d10) time points. Extensive GFP expression *in toto* muscles and in cryo-sections demonstrates that the entire muscle was electroporated. These data have been added in revised Figure 7—figure supplement 1.

8) A physical interaction needs to be demonstrated between b-catenin and SOXF family members to support the authors' conclusions. Also, whether these interactions are lost upon removal of the b-catenin interaction site in the SOXF family needs to be tested.

Physical interaction between SOXF factors and β-catenin has been previously published. Similarly, localization on the β-catenin binding domain on the SOXF proteins sequences, and the loss of interaction with β-catenin following its deletion or point mutation, has been previously reported. We clarified the text and included the appropriate citations:

Chew et al., 2011. SRY-Box Containing Gene 17 Regulates the Wnt/ -Catenin Signaling Pathway in Oligodendrocyte Progenitor Cells." Journal of Neuroscience 31(39): 13921-13935.

Guo et al., L., D. Zhong, S. Lau, X. Liu, X.Y. Dong, X. Sun, V.W. Yang, P.M. Vertino, C.S. Moreno, V. Varma, J.T. Dong, and W. Zhou. 2008.

Kormish JD, Sinner D, Zorn AM. 2010. Interactions between SOX factors and Wnt/β-catenin signaling in development and disease. Dev Dyn. Jan;239(1):56-68. doi: 10.1002/dvdy.22046. Review.

Liu X, Luo M, Xie W, Wells JM, Goodheart MJ, Engelhardt JF. 2010. Sox17 modulates Wnt3A/β-catenin-mediated transcriptional activation of the Lef-1 promoter. Am J Physiol Lung Cell Mol Physiol. Nov;299(5):L694-710. doi: 10.1152/ajplung.00140.2010. Epub 2010 Aug 27.

Sinner et al.,, D., J.J. Kordich, J.R. Spence, R. Opoka, S. Rankin, S.C.J. Lin, D. Jonatan, A.M. Zorn, and J.M. Wells. 2007. Sox17 and Sox4 Differentially Regulate Catenin/T-Cell Factor Activity and Proliferation of Colon Carcinoma Cells. Molecular and Cellular Biology 27:7802-7815.

Sinner et al., D., S. Rankin, M. Lee, and A. Zorn. 2004. Sox17 and catenin cooperate to regulate the transcription of endodermal genes. Development 131:3069-3080.

Zhang, Basta and Klymkowsky, 2005. SOX7 and SOX18 are essential for cardiogenesis in Xenopus. Dev Dyn 234:878-891.

Considering the timing of canonical Wnt activity and SOXF family member expression during regeneration, it is not immediately clear as to the relevance of SoxF factors with b-catenin in the context of this manuscript (Brack et al., 2008; Murphy et al., 2014; Rudolf et al., 2016). Although controversial, b-catenin activity is highest during stages of active fate decisions in satellite cells and myogenic progenitors during regeneration (days 3-7 after injury). Yet in Figure 4, SoxF expression peaks at later stages of regeneration. Also, these transcripts are measured in whole TA muscle, whereas they should be measured in sorted satellite cells and myogenic progenitors.

As requested, satellite cells at 2, 5, and 7 days post-injury were collected with FACS and the expression of different myogenic factors and SoxF genes was evaluated by qRT-PCR. This shows an increase in SoxF expression at d5 and d7, clearly overlapping with days 3-7 after injury. These data have been added in revised Figure 5 and revised Figure 5—figure supplement 1.

Alternatively, SoxF family members along with myogenic fate markers Pax7, MyoD, and Myog could be tested with immunofluorescence at days 3-7 during injury. Since based on the literature and the data in this study, b-catenin activity is associated with myogenic progression, and SoxF family members are proposed to interfere with b-catenin activity; the authors should test localization of SoxF members (Sox17) with myogenic fate regulators at days 3-7 of muscle regeneration. Another possibility is b-catenin activity should be higher in Sox17 null mice, which could explain some of the phenotypes observed in this manuscript (Murphy et al., 2014).

We have tested several SoxF antibodies and have spent a lot of time trying to optimize immunofluorescence with these antibodies on muscle sections. Unfortunately, despite evaluating several protocols, we have not been able to get consistent and reliable results.

9) Figure 6C – β-catenin target gene analyses lack statistical assessment. Also, it appears that the b-cat target genes show variable differences in the Sox17 deficient muscles. The analysis is also complicated by the different cellular and fiber type composition of the muscles in the Sox17 deficient animals. These issues should be accounted for in the authors' presentation and interpretation of these results.

Statistics were added. The manuscript was updated accordingly.

10) Figure 7E Axin2 expression is assayed in sorted Pax3^GFP+^ cells without CD31 negative selection this is problematic based on the authors publications as described above (Goupille et al., 2011; Relaix et al., 2006). Due to the questions with regards to Pax3^GFP^ expression and lack of CD31 use, sort profiles with all gates and populations need to be shown to demonstrate how Pax3^GFP+^ cells were prospectively isolated.

Please see above (answer to comment 1).

11) LiCl is a GSK3b inhibitor, and so, as GSK3b has additional activities that are not related to its role in Wnt signaling, LiCl should not be presented as a specific Wnt activator.

We thank the reviewers for this remark. The text was modified accordingly.

12) The authors need to add an important control to the studies comparing SoxF overexpression and ability to rescue – they must assess the level of overexpression of the various Sox7, Sox17 and Sox18 constructs and ensure that they are similarly overexpressed. Similarly, they provide no evidence that endogenous SOXF protein levels parallel the transcript levels of SoxF genes. These points need to be addressed by the authors.

In the myofiber model used to the rescue experiment (now Figure 7A), it is impossible to directly assess the level of overexpression in the transduced cells due to limiting amount of material. Tagged full-length cDNA constructs were thus generated to avoid variable antibody detection, and transfected in two cell lines. The analysis shows that under the control of the same promoter in the same backbone, expressed protein levels slightly vary between SOXF members at 48h post-transfection and also in a cell to cell fashion (Author response image 3). This observation points not only to intrinsic mRNA and/or protein stability, but also to the cellular system used. Considering the relative levels of the three SOXF factors, we consider that the output of the rescue experiment is unlikely to be significantly affected.

**Author response image 3. respfig3:** Overexpression levels of SOX-FL proteins. C2C12 and HEK293 cells were transfected with GFP-tagged SOXF constructs for 48h. After lysis, 10 µg of proteins were loaded on a 4-12% gradient acrylamide gel. Overexpressed proteins were probed with anti-GFP antibody (Abcam). Loading is controlled using an anti-TBP antibody (Cell Signaling).

13) The authors described an increase of slow fibers in soleus muscle of Sox17^fl^/Pax3^Cre^ mice. First, a more complete analysis of fast and slow myosins should be performed in various muscles should be performed. Second, the authors should at least discuss the potential connection between SOXF expression and muscle metabolism.

Please see above (answer to comment 5).

[Editors' note: further revisions were requested prior to acceptance, as described below.]

The manuscript has been improved but one of the reviewers has raised a few remaining issues that need to be addressed before acceptance, as outlined below:In response to the first review the authors submitted a figure demonstrating lack of Pax3^Cre-GFP^ in CD31^+^ cells. Inclusion of this as a supplement would be helpful.

As suggested by the reviewer, we have added this figure as new Figure 1—figure supplement 1.

It is impressive that the authors observe a similar magnitude of Pax7^+^ SC loss regardless of whether Pax3^Cre^ or Pax7^CreER^ is used. However, there are discrepancies in the regeneration experiments. The Pax3^Cre^-Sox17 KO mice demonstrate obvious impairments in regeneration (Figure 5) after 28 days of recovery. Figure 6 demonstrates Pax7^+^ cell loss in Pax7^CreER^-Sox17 KO mice after only 7 days of recovery. There are no data from 28 day regenerated Pax7^CreER^-Sox17 KO muscle. Therefore it is not known whether regeneration or satellite cell renewal are impaired after 28 days in the Pax7^CreER^-Sox17 KO. These data should be provided and compared/discussed with the Pax3^Cre^ data/published studies to determine if differences in regenerative phenotype occur depending on timing of recombination. Also, some regenerative measures should be quantified for example size of regenerated muscle fibers,% Oil red O area, and% Sirius red area.

As suggested by the reviewers, regeneration after 28 days has been analyzed in the *Sox17*-conditional knockout (*Pax7^CreERT2/+^;Sox17^fl/fl^*) and discussed in the text. These data are included in Figure 6E-L.